# Dynamic control of Hsf1 during heat shock by a chaperone switch and phosphorylation

**Xu Zheng[1][†], Joanna Krakowiak[1][†], Nikit Patel[2], Ali Beyzavi[3][‡], Jideofor Ezike[1], Ahmad S Khalil[2,4]\*, David Pincus[1]\***

[1]Whitehead Institute for Biomedical Research, Cambridge, United States; [2]Department of Biomedical Engineering and Biological Design Center, Boston University, Boston, United States; [3]Department of Mechanical Engineering, Boston University, Boston, United States; [4]Wyss Institute for Biologically Inspired Engineering, Harvard University, Boston, United States

**Abstract** Heat shock factor (Hsf1) regulates the expression of molecular chaperones to maintain protein homeostasis. Despite its central role in stress resistance, disease and aging, the mechanisms that control Hsf1 activity remain unresolved. Here we show that in budding yeast, Hsf1 basally associates with the chaperone Hsp70 and this association is transiently disrupted by heat shock, providing the first evidence that a chaperone repressor directly regulates Hsf1 activity. We develop and experimentally validate a mathematical model of Hsf1 activation by heat shock in which unfolded proteins compete with Hsf1 for binding to Hsp70. Surprisingly, we find that Hsf1 phosphorylation, previously thought to be required for activation, in fact only positively tunes Hsf1 and does so without affecting Hsp70 binding. Our work reveals two uncoupled forms of regulation - an ON/OFF chaperone switch and a tunable phosphorylation gain - that allow Hsf1 to flexibly integrate signals from the proteostasis network and cell signaling pathways.

**\*For correspondence:** akhalil@bu.edu (ASK); pincus@wi.mit.edu (DP)

[†]These authors contributed equally to this work

**Present address:** [‡]David H. Koch Institute for Integrative Cancer Research, Massachusetts Institute of Technology, Cambridge, United States

**Competing interests:** The authors declare that no competing interests exist.

## Introduction

The heat shock response is an ancient and conserved signaling pathway in cells that regulates the expression of molecular chaperones in the presence of thermal and other environmental stresses (*Lindquist, 1986*; *Richter et al., 2010*). Chaperones function to maintain protein homeostasis (proteostasis) by enabling de novo protein folding in the crowded intracellular environment and targeting proteins for degradation (*Dobson, 2003*; *Labbadia and Morimoto, 2015*). Although the heat shock response has been extensively studied, key aspects of the pathway remain a mystery including the mechanisms governing its activation and regulation.

In eukaryotes, the master transcriptional regulator of the heat shock response is heat shock factor 1 (Hsf1) (*Anckar and Sistonen, 2011*). Hsf1 and its cognate DNA binding site, the heat shock element (HSE), represent one of the most conserved protein•DNA interactions known, having been maintained since the last common ancestor of the eukaryotic lineage (*Wu, 1995*). The depth of functional conservation is underscored by the observation that an active form of human Hsf1 can carry out the essential function of Hsf1 in yeast (*Liu et al., 1997*). Indeed, both mammalian and yeast Hsf1 drive a compact set of genes dedicated to proteostasis that forms a densely connected network centered around Hsp70, Hsp40 and Hsp90 (*Mahat et al., 2016*; *Solís et al., 2016*). The small size of the Hsf1 regulon belies its outsize importance in cellular viability.

In addition to maintaining proteostasis at the cellular level, Hsf1 plays important roles in organismal health and disease. Critically, Hsf1 is frequently activated in cancer cells: it has been shown to

**eLife digest** Proteins are strings of amino acids that carry out crucial activities inside cells, such as harvesting energy and generating the building blocks that cells need to grow. In order to carry out their specific roles inside the cell, the proteins need to "fold" into precise three-dimensional shapes.

Protein folding is critical for life, and cells don't leave it up to chance. Cells employ "molecular chaperones" to help proteins to fold properly. However, under some conditions – such as high temperature – proteins are more difficult to fold and the chaperones can become overwhelmed. In these cases, unfolded proteins can pile up in the cell. This leads not only to the cell being unable to work properly, but also to the formation of toxic "aggregates". These aggregates are tangles of unfolded proteins that are hallmarks of many neurodegenerative diseases such as Alzheimer's, Parkinson's and amyotrophic lateral sclerosis (ALS).

Protein aggregates can be triggered by high temperature in a condition termed "heat shock". A sensor named heat shock factor 1 (Hsf1 for short) increases the amount of chaperones following heat shock. But what controls the activity of Hsf1?

To answer this question, Zheng, Krakowiak et al. combined mathematical modelling and experiments in yeast cells. The most important finding is that the 'on/off switch' that controls Hsf1 is based on whether Hsf1 is itself bound to a chaperone. When bound to the chaperone, Hsf1 is turned 'off'; when the chaperone falls off, Hsf1 turns 'on' and makes more chaperones; when there are enough chaperones, they once again bind to Hsf1 and turn it back 'off'. In this way, Hsf1 and the chaperones form a feedback loop that ensures that there are always enough chaperones to keep the cell's proteins folded.

Now that we know how Hsf1 is controlled, can we harness this understanding to tune the activity of Hsf1 without disrupting how the chaperones work? If we can activate Hsf1, we can provide cells with more chaperones. This could be a therapeutic strategy to combat neurodegenerative diseases.

be required for cancer progression in animal models (*Dai et al., 2007*), its activation is associated with poor prognosis in many human cancer patients (*Santagata et al., 2011*), and it drives cancer-specific gene expression programs in both tumor cells and the supporting stroma (*Mendillo et al., 2012*; *Scherz-Shouval et al., 2014*). By contrast, a lack of Hsf1 activity has been suggested to contribute to neurodegenerative diseases with hallmark protein aggregates, and activation of Hsf1 has been proposed as a therapeutic avenue (*Labbadia and Morimoto, 2015*; *Neef et al., 2011*). Moreover, Hsf1 contributes to organismal lifespan (*Hsu et al., 2003*) and protects against obesity (*Ma et al., 2015*).

Despite the deep conservation of Hsf1 and its physiological and clinical importance, the mechanisms regulating Hsf1 activity during stress remain poorly defined, and thus to date it is unclear how Hsf1 controls the heat shock response in cells. Some aspects of Hsf1 regulation are organism- and cell type-specific, such as trimerization, which is a regulated event in mammalian cells but constitutive in yeast (*Sorger et al., 1987*; *Sorger and Nelson, 1989*; *Westwood et al., 1991*). However, two common features are thought to contribute to Hsf1 regulation in all organisms: chaperone titration and phosphorylation (*Anckar and Sistonen, 2011*).

The chaperone titration model suggests that Hsf1 is bound in an inhibitory complex by chaperones in basal conditions (*Voellmy and Boellmann, 2007*). Upon heat shock, the chaperones are titrated away by unfolded or misfolded proteins, leaving Hsf1 free to activate transcription of chaperone genes. Once proteostasis is restored, client-free chaperones again bind to Hsf1 and deactivate it. There is biochemical, pharmacological and genetic evidence to support roles for the Hsp70 and Hsp90 chaperones, their co-chaperones and the TRiC/CCT chaperonin complex in regulating Hsf1 (*Abravaya et al., 1992*; *Baler et al., 1992*, *1996*; *Duina et al., 1998*; *Guo et al., 2001*; *Neef et al., 2014*; *Ohama et al., 2016*; *Shi et al., 1998*; *Zou et al., 1998*). However, direct, unequivocal evidence for this model – i.e., a complete cycle of Hsf1 'switching' by dynamic dissociation and re-association with specific chaperone(s) during heat shock – is lacking. As a result, though widely invoked, the details of the chaperone titration model remain unclear.

The second putative Hsf1 regulatory mechanism common across organisms is heat shock-dependent phosphorylation. Multiple phosphorylation sites have been mapped on Hsf1 (*Anckar and Sistonen, 2011*; *Guettouche et al., 2005*), and mutational analysis has suggested activating, repressing, fine-tuning and condition-specific roles for individual sites of phosphorylation in yeast and mammalian cells (*Budzynski et al., 2015*; *Cho et al., 2014*; *Dai et al., 2015*; *Hahn and Thiele, 2004*; *Hashikawa et al., 2006*; *Hashikawa and Sakurai, 2004*; *Hietakangas et al., 2003*; *Høj and Jakobsen, 1994*; *Holmberg et al., 2001*; *Kline and Morimoto, 1997*; *Knauf et al., 1996*; *Lee et al., 2013*; *Soncin et al., 2003*; *Sorger and Pelham, 1988*; *Tang et al., 2015*; *Wang et al., 2003*; *Yamamoto et al., 2007*). Recent work in which 15 phosphorylation sites were simultaneously mutated in human Hsf1 failed to disrupt Hsf1 activation during heat shock (*Budzynski et al., 2015*), and a genome-wide RNAi-based screen for modulators of Hsf1 activity found no evidence for kinase regulation (*Raychaudhuri et al., 2014*). Thus, despite being a hallmark of the heat shock response, no clear role for Hsf1 phosphorylation has yet emerged.

Here, we combine experimental and theoretical approaches to elucidate the mechanism of Hsf1-mediated activation and control of the heat shock response in budding yeast. Specifically, we combine mass spectrometry, biochemistry, mathematical modeling, genetics and synthetic biology to propose and validate that Hsp70 dynamically interacts with Hsf1 to form the basis of a bona fide activation 'switch' and feedback loop that regulates Hsf1 activity during heat shock. Based on this finding, we then investigate the role and quantitative contribution of Hsf1 phosphorylation in regulating the output of the heat shock transcriptional response. We use en masse mutational analysis, combined with transcriptomic and genome-wide ChIP-seq measurements, to systematically define the function of Hsf1 phosphorylation. We find that phosphorylation is fully dispensable for Hsf1 activation during heat shock, but contributes by enhancing transcriptional output levels as a 'positive gain'. Phosphorylation does not control the interaction between Hsf1 and Hsp70, but rather enhances transcription independent of whether Hsp70 is bound or not. Our findings reveal that Hsf1 uses these two modes of regulation – a chaperone switch and phosphorylation fine-tuning – in a largely uncoupled fashion to dynamically control the heat shock response. We propose that this allows Hsf1 to integrate diverse signaling information without disrupting its direct readout of the proteostasis network.

## Results

### Hsp70 binds to Hsf1 and transiently dissociates during heat shock

To identify proteins involved in the dynamic regulation of Hsf1 during heat shock, we performed immunoprecipitation (IP) of Hsf1 from cells harvested over time following a shift from 25°C to 39°C and analyzed the IP samples by mass spectrometry (MS). We expressed dual epitope-tagged Hsf1-3xFLAG-V5 as the only copy of Hsf1 from its endogenous promoter and performed serial affinity purifications to reduce non-specific binding (*Figure 1—figure supplement 1A*, see Materials and methods). MS analysis revealed that the only proteins that co-precipitated with Hsf1 in basal conditions in each of three independent replicates were the cytosolic Hsp70 chaperones Ssa1 and Ssa2 (jointly Ssa1/2) which share 98% sequence identity (*Figure 1—figure supplement 1B,C*, *Figure 1—source data 1*). No peptides derived from Ssa1/2 were identified in an untagged control or when we expressed YFP-3xFLAG-V5 and purified it following the same protocol (*Figure 1—source data 1*). Notably, no peptides belonging to Hsp90 or any other chaperones were identified in any of the samples. Intriguingly, following a five-minute heat shock, Ssa1/2 were absent in two IP replicates and were greatly diminished from the third replicate (*Figure 1—figure supplement 1B,C*, *Figure 1—source data 1*). No proteins co-precipitated with Hsf1 exclusively in heat shock conditions. Western blot analysis of Hsf1 IP samples collected over a heat shock time course revealed a transient decrease in the relative amount of Ssa1/2 followed by restoration of Ssa1/2 levels at the later time points (*Figure 1A*).

To validate that Hsf1 and Hsp70 directly interact, we tested their ability to bind to each other in vitro. We purified recombinant Hsf1-6xHIS and 6xHIS-3xFLAG-Ssa2 from *E. coli*, mixed the recombinant proteins together at an equimolar ratio and performed an anti-FLAG IP. We precipitated Hsf1 only in the presence of Ssa2, demonstrating that they can specifically and directly bind to each other (*Figure 1—figure supplement 1D*). We reversed the affinity tags and likewise precipitated Ssa2 in

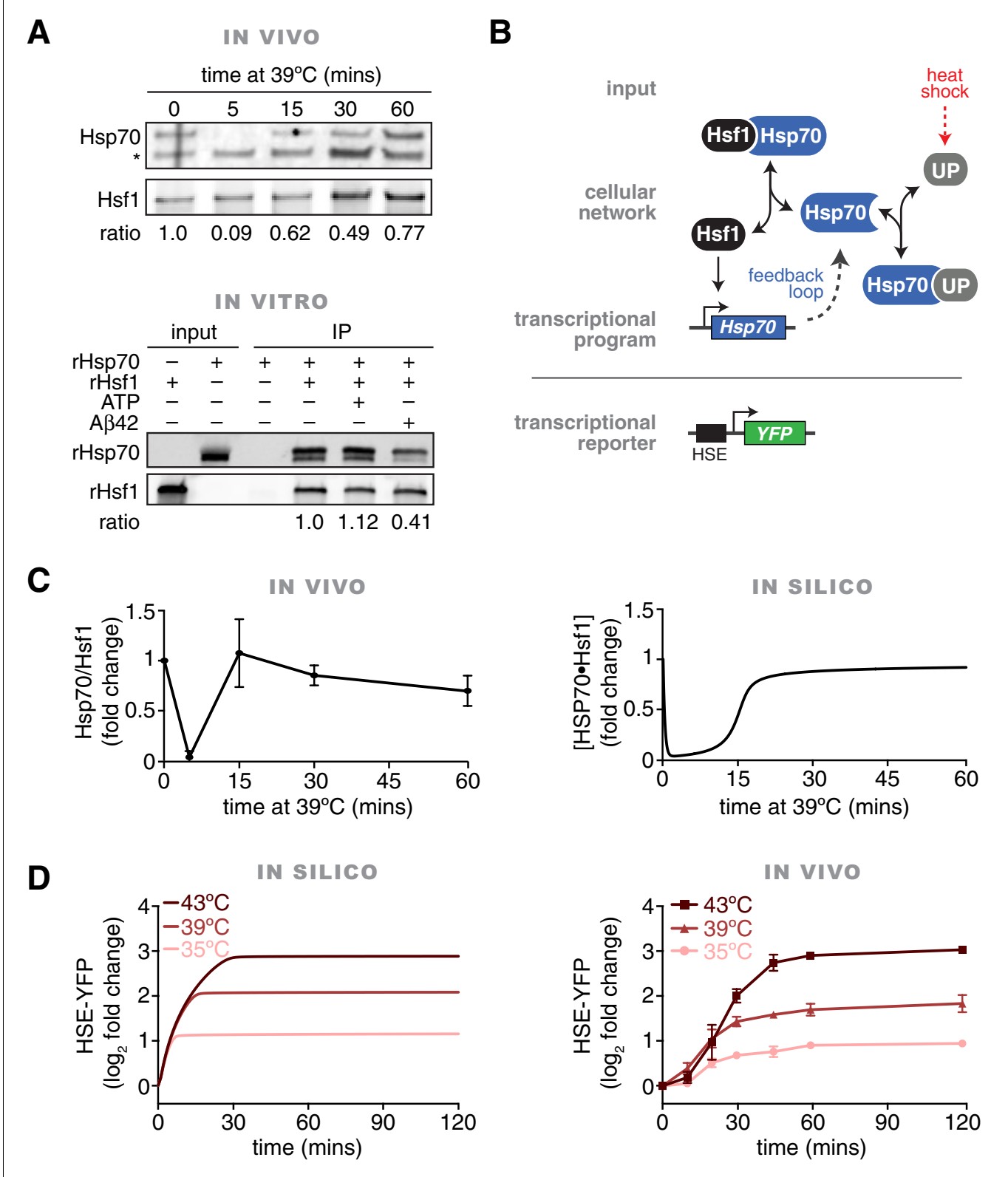

**Figure 1.** In vivo, in vitro and in silico evidence for an Hsp70•Hsf1 dissociation switch as the core mechanism regulating the heat shock response. (**A**) IP/ Western blot showing Hsp70 transiently dissociating from Hsf1 during heat shock (upper panel). Western blots were probed with antisera recognizing Ssa1/2 (top) and an anti-FLAG antibody to recognize Hsf1 (bottom). IP of recombinant proteins were performed with rHsf1-3xFLAG as bait and analyzed by Western blot (lower panel). Blots were probed with an anti-HIS antibody to recognize recombinant Ssa2 (rSsa2, top) and with an anti-FLAG antibody

*Figure 1 continued on next page*

*Figure 1 continued*

to recognize recombinant Hsf1 (rHsf1, bottom). The IPs were also performed in the presence of 1 mM ATP or five-fold molar excess of Aβ42 peptide. The numbers below the blots indicate the normalized ratio of Ssa2/Hsf1. (B) Cartoon schematic of the mathematical model of Hsf1 regulation illustrating the network connections and the feedback loop. UP is an abbreviation for 'unfolded proteins'. See *Figure 1—figure supplement 2* and Materials and methods for details, equations and parameters. (C) Quantification of the top three peptides derived from Hsp70 proteins Ssa1 or Ssa2 (Ssa1/2 are grouped due to 98% identity) relative to the top three peptides from Hsf1 as determined by IP/MS (left panel). The IP experiments were performed in triplicate at the indicated time points following a shift to 39°C. See *Figure 1—figure supplement 1*. The values are the average of the three replicates and error bars depict the standard deviation. Source data are included as *Figure 1—source data 1*. Simulation of the levels of the Hsf1•Hsp70 complex over time following a shift from 25°C to 39°C (right panel). (D) Simulation of the levels of the HSE-YFP reporter over time following upshift from 25°C to the indicated temperatures (left panel). Flow cytometry measurements of cells expressing the HSE-YFP reporter following upshift from 25°C to the indicated temperatures (right panel). See Materials and methods for assay and analysis details.

The following source data and figure supplements are available for figure 1:

**Source data 1.** Table of peptide counts from proteins identified in Hsf1-3xFLAG-V5 IP/MS experiments.
**Figure supplement 1.** Ssa2 co-precipitates with Hsf1 in basal conditions but is greatly reduced immediately following heat shock.
**Figure supplement 2.** Description and parameterization of mathematical model of the heat shock response.

the presence of 3xFLAG-Hsf1 (*Figure 1A*). Addition of ATP neither enhanced nor disrupted the interaction between Hsf1 and Ssa2 (*Figure 1A*). By contrast, addition of a five-fold molar excess of the aggregation-prone, Alzheimer's disease-associated Aβ42 peptide reduced the interaction between Hsf1 and Ssa2 (*Figure 1A*). This suggests that hydrophobic peptides can titrate Hsp70 away from Hsf1. Taken together, these in vivo and in vitro results confirm that Hsp70 dynamically dissociates and re-associates with Hsf1 during heat shock.

## A mathematical model of the heat shock response

Based on our finding that Hsp70 transiently dissociates from Hsf1 during heat shock, we next sought to develop a simple mathematical model of the heat shock response. The goal of the model was two-fold: (1) to quantitatively explore if Hsp70 interaction 'switching' could serve as a core, minimal regulatory mechanism that recapitulates the dynamics of heat shock response; (2) to generate quantitative predictions to further test this molecular mechanism. Our model consisted of a system of six coupled ordinary differential equations, describing a feedback loop in which free Hsf1 induces production of Hsp70, and free Hsp70 in turn binds to Hsf1 in a transcriptionally inactive complex (*Figure 1B*, *Figure 1—figure supplement 2A*, see Materials and methods). In addition to binding to Hsf1, Hsp70 can also bind to an unfolded protein (UP), but cannot bind to both Hsf1 and UP simultaneously. UPs are introduced at a level that is based on the temperature in accordance with biophysical measurements (*Figure 1—figure supplement 2B*, see Materials and methods) (*Lepock et al., 1993*; *Scheff et al., 2015*). In this manner, temperature upshifts increase the level of UPs, simulating protein folding stress caused by heat shock. UPs titrate Hsp70 away from Hsf1, leaving Hsf1 free to activate expression of more Hsp70, thus generating the feedback loop. This minimal model ignores any potential role for phosphorylation or other post-translational modifications in regulating Hsf1 activity.

We used the apparent kinetics of Hsp70 dissociation from Hsf1 (*Figure 1C*) to constrain the model by computationally screening for parameter sets that captured key features of these data (*Figure 1—figure supplement 2C*, see Materials and methods) (*Ma et al., 2009*). This parameter screening approach revealed that a broad range of values satisfied the experimental constraints (*Figure 1—figure supplement 2C*). We settled on parameter values that appeared most frequently in simulations that fulfilled the experimental constraints (*Figure 1C*).

## The chaperone titration model recapitulates Hsf1 activation dynamics

We tested the mathematical model by simulating time courses of temperature upshifts (25°C to 35°C, 25°C to 39°C, and 25°C to 43°C). As an output of the model, we tracked transcription of a 'reporter' whose levels are dependent on Hsf1 activity. This in silico reporter can be directly

compared to experimental measurements of an Hsf1-dependent HSE-YFP reporter that we integrated into the yeast genome (*Brandman et al., 2012*) (*Figure 1B*). Simulations of the HSE-YFP reporter generated induction curves that reached temperature-dependent plateaus (*Figure 1D*). Since YFP is a long-lived protein – in cells and in the model – the plateaus indicate that no more YFP is being produced in the simulation and Hsf1 has deactivated. We performed the same temperature step time course experiments in cells and measured the HSE-YFP reporter at discrete time points by flow cytometry. As predicted by the model, we observed induction leading to temperature-dependent maximal responses (*Figure 1D*). However, following the initial rapid accumulation of YFP that was predicted by the model, the cells continued to accumulate signal slowly through the later time points (this later phase of transcriptional output is addressed further below). These results suggest that a chaperone titration model is sufficient to account for the immediate Hsf1 activation dynamics in response to temperature steps.

## Predicting synthetic perturbations of the Hsf1-Hsp70 feedback loop

To rigorously test our model of the heat shock response, we used the mathematical framework to predict the Hsf1 activation response for synthetic perturbations of the feedback loop. For example, in addition to activation by UPs (via increases in temperature), the model predicted that Hsf1 transcriptional activity can be driven by Hsf1 overexpression in the absence of temperature upshifts (*Figure 2A,C*). Less intuitively, the model also predicted that overexpression of an Hsf1 'decoy', which lacks the ability to trimerize or bind DNA, would activate endogenous Hsf1 by titrating Hsp70 (*Figure 2B,C*). We next used synthetic biology approaches to experimentally test the model predictions. We constructed a decoy mutant by removing the DNA binding and trimerization domains from Hsf1 and replacing them with the well-folded fluorescent protein mKate2 (*Figure 2B*). We then placed either a wild type allele of *HSF1* or the decoy under the synthetic control of a $\beta$-estradiol (estradiol) inducible system (*Pincus et al., 2014*) (*Figure 2A,B*, *Figure 2—figure supplement 1A*, see Materials and methods). In agreement with the model, both full length Hsf1 and the decoy activated the HSE-YFP reporter in dose-dependent manners, with full length Hsf1 serving as a more potent activator than the decoy (*Figure 2D*). By contrast, expression of mKate2 alone led to only a modest increase in HSE-YFP levels as a function of estradiol (*Figure 2D*). Importantly, the decoy does not form aggregates when overexpressed, remaining diffusely localized in the nucleus in both basal and heat shock conditions, but does interact with the Hsp70 chaperones Ssa1/2 and disrupts the interaction between endogenous Hsf1 and Ssa1/2 in co-IP assays (*Figure 2—figure supplement 1B*, *Figure 2—source data 1*). Domain truncation analysis of the decoy revealed that the C-terminal activation domain of Hsf1 is both necessary and sufficient to activate endogenous Hsf1, while the N-terminal activation domain is dispensable (*Figure 2—figure supplement 1C,D*). Thus, the decoy is not merely a UP, but rather functions as a specific activator of Hsf1, likely by titrating away Hsp70 via its C-terminal activation domain.

## Hsf1 overexpression inhibits cell growth

While overexpression of either full length Hsf1 or the decoy activated the HSE-YFP reporter, neither was innocuous: both inhibited cell growth in a dose-dependent manner. Full length Hsf1 impaired growth 20-fold more than the decoy, and the decoy impaired growth three-fold more than mKate2 alone at the highest dose of estradiol (*Figure 3A*). The growth impairment caused by Hsf1 overexpression was not the result of a specific cell cycle arrest, as the remaining cells displayed asynchronous cell cycle stages (*Figure 3—figure supplement 1A,B*).

Since both full length Hsf1 and the decoy bound to Hsp70 and induced the transcriptional response, the growth inhibition could be due to either Hsp70 sequestration or the transcriptional induction itself through 'squelching' of the transcriptional machinery (*Gill and Ptashne, 1988*) and/ or gratuitous gene expression. To isolate the consequences of inducing the transcriptional program in the absence of stress, we constructed a synthetic fusion of the Hsf1 DNA binding and trimerization domains with a transcriptional activation domain derived from the herpes simplex virus protein 16 (DBD$_{Hsf1}$-VP16) (*Sadowski et al., 1988*), and placed this fusion under estradiol control. DBD$_{Hsf1}$-VP16 was a more potent inducer of the HSE-YFP reporter than full length Hsf1 but impaired growth equally to full length Hsf1 across the estradiol dose response (*Figure 3—figure supplement 1C,D*). These data suggest that over-activating the Hsf1 transcriptional program impairs growth.

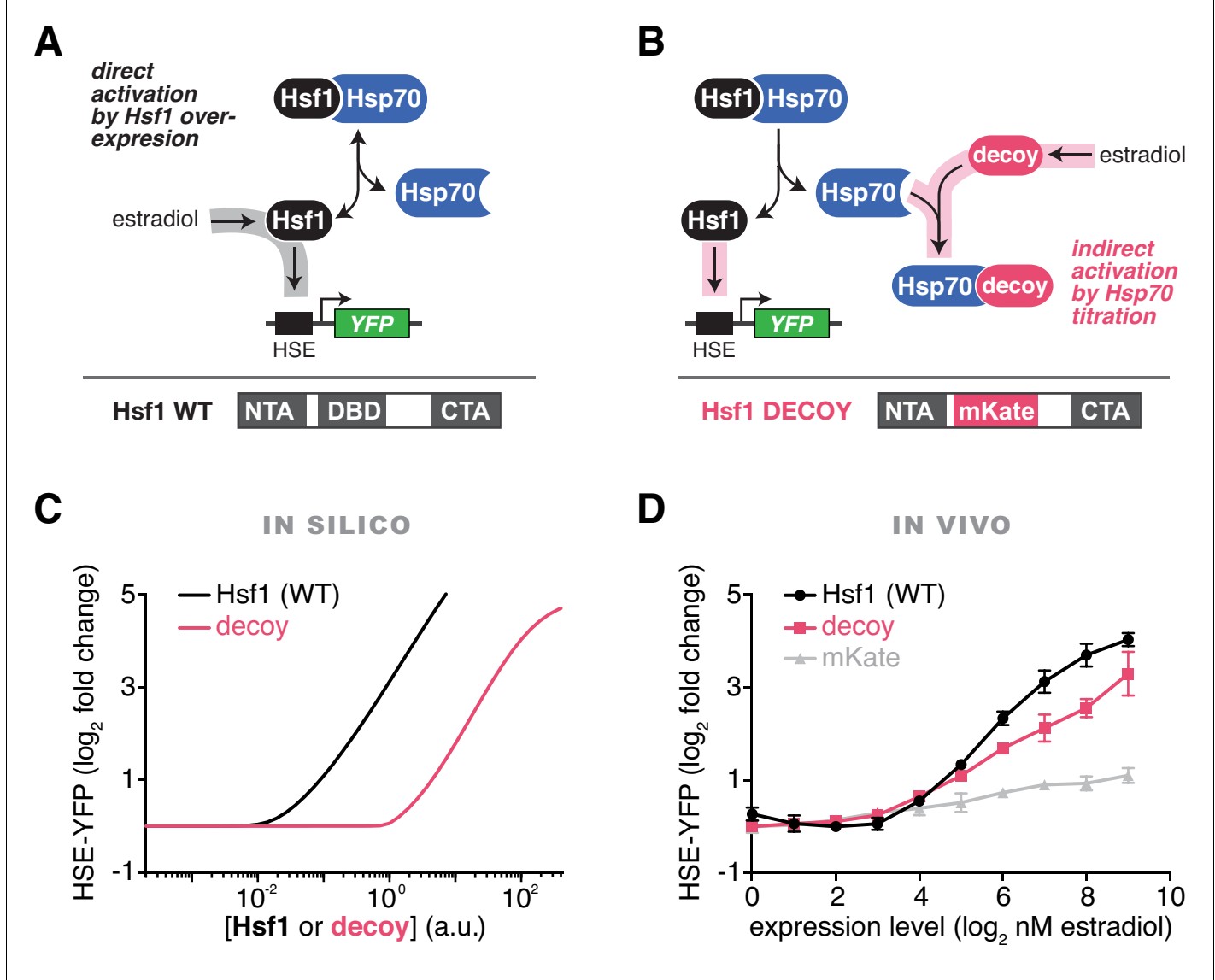

**Figure 2.** Prediction and validation of synthetic perturbations to the Hsf1-Hsp70 feedback loop. (**A**) Cartoon schematic of activation by overexpressing full length Hsf1. Hsf1 can be expressed at many different levels by titrating the concentration of estradiol in the media (See *Figure 2—figure supplement 1* and Materials and methods). The Hsf1 domain architecture is displayed below. The DNA binding domain (DBD) is between N- and C-terminal activation domains (NTA and CTA). (**B**) Cartoon schematic of activation via overexpression of the Hsf1 decoy. The decoy domain architecture is displayed below. (**C**) Simulation of the HSE-YFP reporter as a function of the expression level of full length Hsf1 or the decoy. (**D**) Experimental measurement of the HSE-YFP reporter by flow cytometry in cells expressing full length Hsf1, the decoy or mKate alone across a dose response of estradiol. Cells were monitored following growth in the presence of the indicated concentrations of estradiol for 18 hr. Data points are the average of median YFP values for three biological replicates, and error bars are the standard deviation. See Materials and methods for assay and analysis details.

The following source data and figure supplement are available for figure 2:

**Source data 1.** Table of peptide counts from proteins identified in decoy IP/MS experiments with decoy-3xFLAG-V5 and Hsf1-3xFLAG-V5 as bait.

**Figure supplement 1.** Overexpression of a decoy of Hsf1 activates endogenous Hsf1.

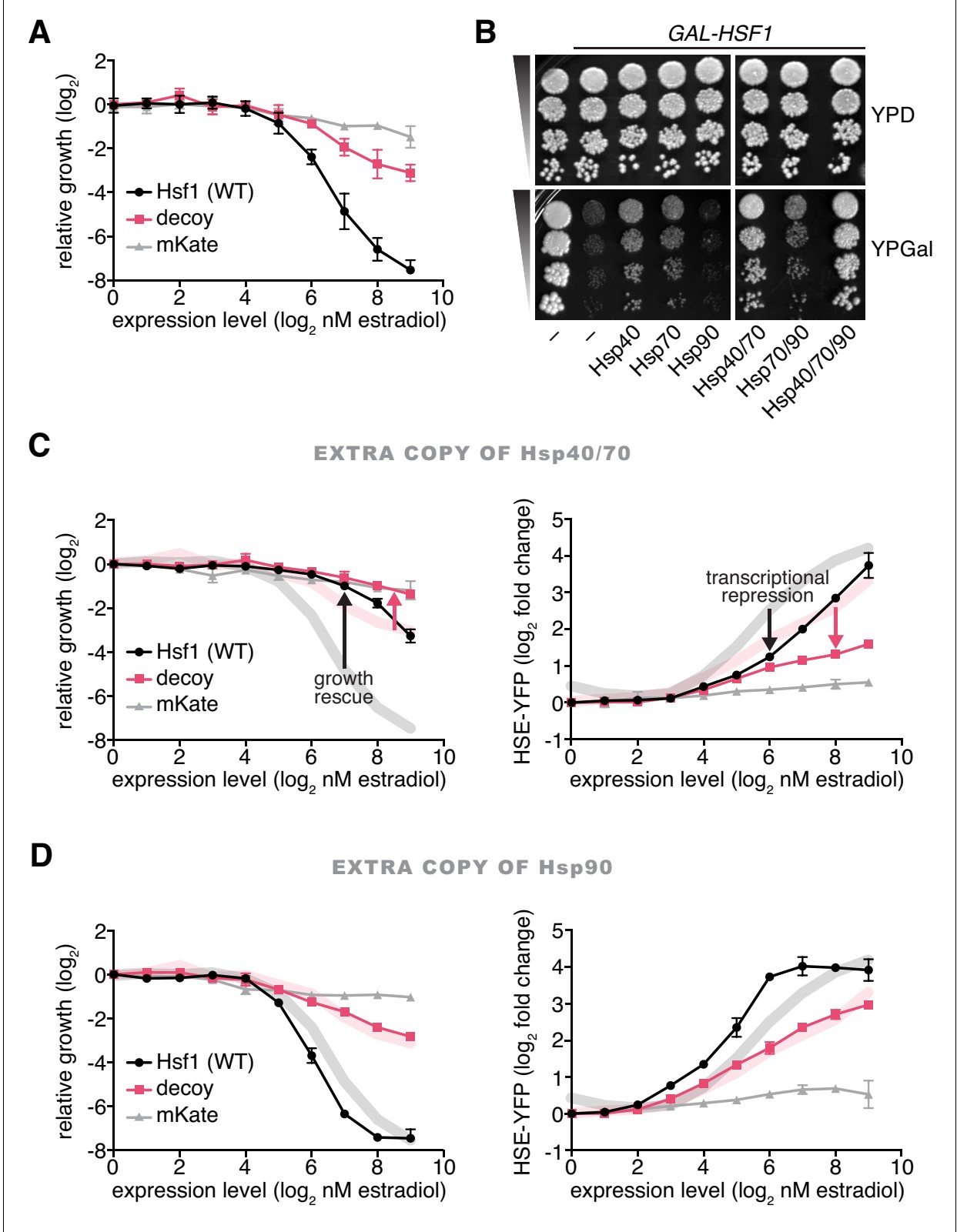

**Figure 3.** Hsp70 and Hsp40 suppress Hsf1 overexpression. (**A**) Cells expressing full length Hsf1, the decoy or mKate alone were assayed for growth by flow cytometry following 18 hr of incubation with the indicated doses of estradiol. Data points are the average of normalized cell count values for three biological replicates, and error bars are the standard deviation. See Materials and methods for assay and analysis details. (**B**) Dilution series spot assays in the absence and presence of galactose to monitor growth of cells expressing full length Hsf1 from the *GAL1* promoter. Ydj1 (Hsp40), Ssa2 (Hsp70),

*Figure 3 continued on next page*

*Figure 3 continued*

Hsc82 (Hsp90) and combinations thereof were expressed from strong Hsf1-independent promoters and assayed for their ability to rescue the growth defect caused by Hsf1 overexpression. (C) Cells expressing an extra copy of Hsp70 and Hsp40 were assayed for growth (left panel) and for induction of the HSE-YFP reporter (right panel) as a function of the expression level of full length Hsf1, the decoy or mKate by flow cytometry following 18 hr of incubation with the indicated doses of estradiol. The thick lines in the background are the reference curves for cells lacking extra chaperone expression (taken from *Figures 3A* and *2D*). (D) Cells expressing an extra copy of Hsp90 were assayed for growth (left panel) and for induction of the HSE-YFP reporter (right panel). Reference curves are depicted as above.

The following figure supplements are available for figure 3:

**Figure supplement 1.** Hsf1 overexpression does not lead to a specific cell cycle arrest and cannot be explained by induction of a gratuitous transcriptional program.

**Figure supplement 2.** The Hsp40 Ydj1 is not required for the interaction between Hsf1 and Hsp70.

## Hsp70 and Hsp40 suppress Hsf1 overexpression

Since Hsf1 overexpression impairs growth, inhibitors of Hsf1 activity should be genetic suppressors of the growth phenotype. To test if chaperones would behave as Hsf1 inhibitors, we placed Ssa2 (Hsp70), Hsc82 (Hsp90) and Ydj1 (Hsp40) under the control of strong Hsf1-independent promoters (see Materials and methods) (*Solís et al., 2016*) and assayed for their ability to suppress the growth defect of cells overexpressing Hsf1. Interestingly, both Hsp70 and Hsp40 partially suppressed the growth inhibition caused by Hsf1 overexpression, while Hsp90 failed to provide any growth rescue (*Figure 3B*). These data are consistent with prior reports in mammalian cells showing that overexpression of Hsp70 and Hsp40 attenuate Hsf1 activity (*Shi et al., 1998*) and in *C. elegans* showing that loss of Hsp70 results in >10 fold more Hsf1 activation than loss of Hsp90 (*Guisbert et al., 2013*). Since Hsp40 chaperones deliver substrates and stimulate the ATPase activity of Hsp70 chaperones (*Kampinga and Craig, 2010*), Hsp40 may be enhancing the activity of endogenous Hsp70 to suppress Hsf1 overexpression. To determine if Hsp40 is required for efficient Hsp70 binding to Hsf1, we deleted *YDJ1* and performed a co-IP heat shock time course. We observed that Hsp70 was still able to robustly bind to Hsf1 under basal conditions and transiently dissociate during heat shock (*Figure 3—figure supplement 2A*). However, basal Hsf1 activity was increased >3 fold as measured by the HSE-YFP reporter in *ydj1∆* cells (*Figure 3—figure supplement 2B*), indicating that there is likely to be greater total Hsp70 in these cells and complicating direct comparison of *ydj1∆* and wild type cells. Thus, overexpression of Hsp40 may directly increase the ability for Hsp70 to bind to Hsf1 or generally improve global proteostasis such that there is more unoccupied Hsp70 available to repress Hsf1. Co-expression of Hsp70 and Hsp40 afforded more rescue than either chaperone alone, but addition of Hsp90 did not enhance the suppression provided by Hsp70 alone (*Figure 3B*). Hsp70 and Hsp40 together diminished the ability of Hsf1 and the decoy to inhibit growth and induce the HSE-YFP reporter across the estradiol dose response (*Figure 3C*). By contrast, Hsp90 failed to rescue growth or reduce HSE-YFP induction (*Figure 3D*). Taken together, these data support a model in which Hsp70, assisted by Hsp40, represses Hsf1 with little contribution from Hsp90.

## Hsf1 can be phosphorylated on 73 sites

The mathematical model along with the biochemical and genetic data implicates Hsp70 as the predominant regulator of Hsf1. Given these results, we wondered if we could identify a role for phosphorylation in regulating Hsf1 activity. To address this question, we mapped Hsf1 phosphorylation sites in basal and heat shock conditions, performed site-directed mutagenesis of identified sites and assayed for changes in HSE-YFP levels (see Materials and methods). Across 11 IP/MS experiments, we observed phosphorylation of 73 out of 153 total serine and threonine (S/T) residues in Hsf1 in at least one condition (*Figure 4—source data 1*). Strikingly, none of the 40 single point mutations we tested – whether mutated to alanine to remove the ability to be phosphorylated or to aspartate to mimic phosphorylation – showed significant differences in HSE-YFP levels compared to wild type Hsf1 in basal or heat shock conditions, even when we created sextuple mutants of clustered residues (*Figure 4—figure supplement 1A*).

## Removing phosphorylation modestly reduces Hsf1 activity during heat shock

Since Hsf1 activity was robust to so many mutations, we opted to remove phosphorylation altogether. We generated a synthetic *HSF1* allele with all 153 S/T codons mutated to alanine. We chose to mutate all S/T codons rather than just the 73 identified phosphorylation sites in order to prevent utilization of alternative sites, ensuring that we completely removed the ability to be phosphorylated. *HSF1* is an essential gene in yeast, and when we expressed this mutant as the only copy of *HSF1*, unsurprisingly the cells were inviable. However, restoration of a single conserved serine (S225), which we found was required for Hsf1 to bind DNA in vitro, was sufficient to restore growth ( *Figure 4—figure supplement 1B,C*). While wild type Hsf1 robustly incorporated $^{32}$P during heat shock, this 152 alanine-substituted mutant (termed Hsf1$^{\Delta po4}$) showed no $^{32}$P incorporation during heat shock though it was stably expressed (*Figure 4A*). Despite its 152 mutations, Hsf1$^{\Delta po4}$ showed normal subcellular localization and unaltered DNA binding across the genome (*Figure 4—figure supplement 1D,E*).

In addition to supporting growth in basal conditions, Hsf1$^{\Delta po4}$ allowed cells to grow at elevated temperature, albeit with a deficit compared to wild type cells (*Figure 4B*). In accordance, Hsf1$^{\Delta po4}$ induced the HSE-YFP reporter in response to heat shock to 75% of the level induced by wild type Hsf1 (*Figure 4C*). Global analysis of mRNA levels by deep sequencing (RNA-seq) revealed that wild type and Hsf1$^{\Delta po4}$ cells were highly correlated across the transcriptome in basal conditions ($R^2$ = 0.98, *Figure 4D*, *Figure 4—source data 2*). However, while the correlation remained robust in response to heat shock ($R^2$ = 0.86), the subset of genes that most strongly depend on Hsf1 for their transcription (HDGs, for Hsf1-dependent genes) (*Solís et al., 2016*) fell below the correlation axis (*Figure 4E*, *Figure 4—source data 2*). These data suggest that Hsf1 phosphorylation is required for its full potency as a transcriptional activator.

## Mimicking constitutive hyper-phosphorylation activates Hsf1 in basal conditions

Since phosphorylation appeared to be necessary to fully activate Hsf1 during heat shock, we wondered if it would be sufficient to drive increased transcription in the absence of stress. To test this, we generated a synthetic mutant, termed Hsf1$^{PO4*}$, in which all 116 S/T residues that fall outside of the DNA binding and trimerization domains were replaced with aspartate to mimic constitutive hyper-phosphorylation (*Figure 4A*). Indeed, when expressed as the only copy of Hsf1 in the cell, Hsf1$^{PO4*}$ activated the HSE-YFP reporter to a level seven-fold higher than wild type Hsf1 in basal conditions (*Figure 4C*). Despite its high basal activity, Hsf1$^{PO4*}$ was able to induce the HSE-YFP even further during heat shock, suggesting its activity is still restrained by Hsp70 binding in basal conditions (*Figure 4C*). RNA-seq analysis corroborated the hyperactivity of Hsf1$^{PO4*}$ in basal conditions as well as its heat shock inducibility (*Figure 4F,G*). However, while Hsf1$^{PO4*}$ cells grew comparably to wild type cells at 30°C, they showed marked growth inhibition at 37°C (*Figure 4B*). Thus, rather than protecting cells against proteotoxic stress, hyper-activation of Hsf1 impaired growth during stress.

Since Hsf1$^{PO4*}$ contained 116 aspartate substitutions, we wondered if bulk negative charge was driving enhanced transcriptional activation. To test if the number of negatively charged residues would be proportional to transcriptional activity, we generated mutants with 33, 49 or 82 aspartate substitutions in an otherwise Hsf1$^{\Delta po4}$ background (*Figure 4H*). Thus, there is no opportunity for heat shock-induced phosphorylation. Remarkably, we observed a direct correspondence between total negative charge and the transcriptional output (*Figure 4I*). These data suggest that increasing Hsf1 phosphorylation positively tunes its transcriptional activity.

## Hsf1 phosphorylation does not interfere with its interaction with Hsp70

The observation that Hsf1$^{PO4*}$ retained its heat shock inducibility suggested that phosphorylation plays little or no role in its regulation by Hsp70. To test this, we deployed the decoy activation assay described above (*Figure 2B*). Since activation of Hsf1 by the decoy depends on titration of Hsp70 away from endogenous Hsf1, we wondered if the phosphorylation state of the decoy would affect its ability to induce the HSE-YFP reporter. We generated additional decoy mutants derived from Hsf1$^{\Delta po4}$ and Hsf1$^{PO4*}$ to remove or mimic constitutive phosphorylation (*Figure 5A*). Like the wild

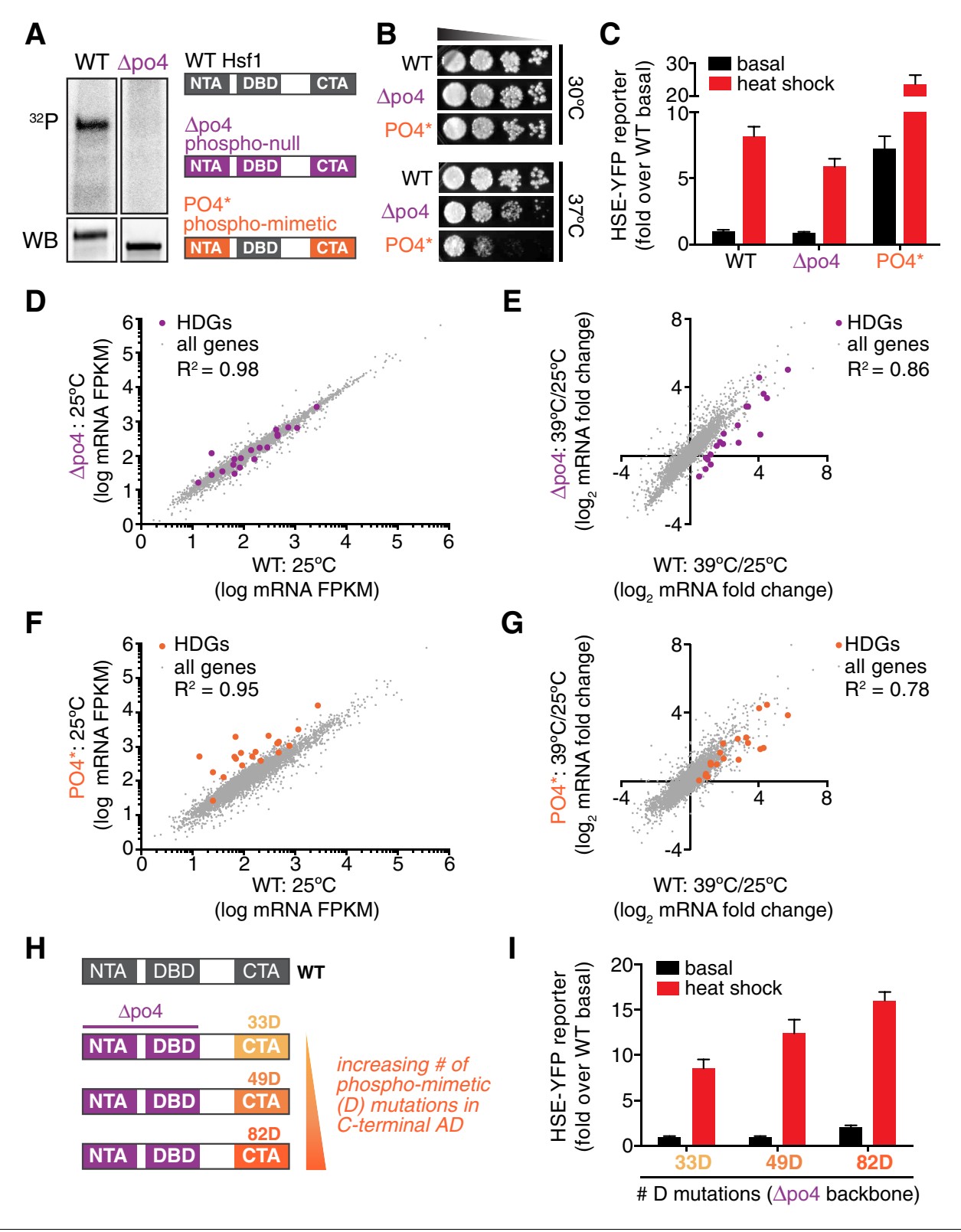

**Figure 4.** Phosphorylation is dispensable for Hsf1 function but tunes the gain of its transcriptional activity. (A) $^{32}$P incorporated into wild type Hsf1-3xFLAG-V5 and Hsf1$^{\Delta po4}$-3xFLAG-V5 during 30 min of heat shock (upper panel). Hsf1-3xFLAG-V5 and Hsf1$^{\Delta po4}$-3xFLAG-V5 were affinity purified by anti-FLAG IP, resolved by SDS-PAGE and phosphor-imaged. Western blot of total lysate from wild type Hsf1-FLAG-V5 and Hsf1$^{\Delta po4}$-FLAG-V5 cells was probed with an anti-FLAG antibody; Hsf1$^{\Delta po4}$-FLAG migrates faster than wild type Hsf1-FLAG (lower panel). Schematics of the domain architecture and

*Figure 4 continued on next page*

Zheng *et al.* eLife 2016;5:e18638. DOI: 10.7554/eLife.18638

Figure 4 continued

color code for wild type Hsf1, Hsf1$^{\Delta po4}$ and Hsf1$^{PO4*}$. (B) Wild type, Hsf1$^{\Delta po4}$ and Hsf1$^{PO4*}$ cells were monitored for growth by dilution series spot assays. Cells were incubated at the indicated temperatures for two days. (C) Wild type, Hsf1$^{\Delta po4}$ and Hsf1$^{PO4*}$ cells expressing the HSE-YFP reporter were assayed for Hsf1 transcriptional activity in control and heat shock conditions by flow cytometry. Bars are the average of median YFP values for three biological replicates, and error bars are the standard deviation. See Materials and methods for assay and analysis details. (D) Genome-wide mRNA levels were quantified in basal conditions in wild type and Hsf1$^{\Delta po4}$ cells by RNA-seq. Within each sample, relative expression levels for each mRNA (gray dots) are plotted as fragments per kilobase per million mapped reads (FPKM). Hsf1-dependent genes (HDGs) are highlighted in purple. Source data are included as **Figure 4—source data 2**. (E) Fold changes of each mRNA in heat shock conditions compared to basal conditions were calculated for wild type and Hsf1$^{\Delta po4}$ cells and plotted against each other. Hsf1-dependent genes (HDGs) are highlighted in purple. Source data are included as **Figure 4—source data 2**. (F) Genome-wide mRNA levels were quantified in basal conditions in wild type and Hsf1$^{PO4*}$ cells by RNA-seq (gray dots). Hsf1-dependent genes (HDGs) are highlighted in orange. Source data are included as **Figure 4—source data 2**. (G) Fold changes of each mRNA in heat shock conditions compared to basal conditions were calculated for wild type and Hsf1$^{PO4*}$ cells and plotted against each other. Hsf1-dependent genes (HDGs) are highlighted in orange. Source data are included as **Figure 4—source data 2**. (H) Schematic of mutants with different numbers of aspartate (D) residues. 33, 49 or 82 D residues were introduced in the CTA in the Δpo4 background. (I) Mutants depicted in (H) expressing the HSE-YFP reporter were assayed for Hsf1 transcriptional activity in control and heat shock conditions by flow cytometry as above.

The following source data and figure supplement are available for figure 4:

**Source data 1.** Table of Hsf1 phosphorylation sites identified in Hsf1-3xFLAG-V5 IP/MS IP/MS experiments in various conditions.
**Source data 2.** Table of genome wide transcript levels as measured by RNA-seq under basal (30°C) and heat shock conditions (30 min at 39°C) in wild type, Hsf1$^{\Delta po4}$ and Hsf1$^{PO4*}$ cells.
**Figure supplement 1.** Mutational analysis of Hsf1 phosphorylation sites reveals a single essential serine and allows for generation of a phospho-mimetic.

type decoy, these constructs contained mKate2 in place of the DNA binding and trimerization domains and were expressed across a dose response of estradiol. The wild type, Δpo4 and PO4* decoys displayed superimposable HSE-YFP induction profiles with matching slopes as a function of absolute expression level (**Figure 5B**). Thus, neither removing nor mimicking phosphorylation altered the activity of the decoy, suggesting that phosphorylation does not affect the ability of Hsf1 to bind to Hsp70.

To directly test whether Hsf1 phosphorylation modulates its interaction with Hsp70, we phosphorylated Hsf1 in vitro with purified casein kinase II (CKII) and monitored its binding to 3xFLAG-Ssa2. CKII phosphorylated Hsf1as evidenced by its mobility shift, yet 3xFLAG-Ssa2 retained the ability to pull down Hsf1 in an anti-FLAG IP comparably to non-phosphorylated controls (**Figure 5C**). These data indicate that phosphorylation does not preclude binding between Hsf1 and Hsp70.

## Mimicking hyper-phosphorylation increases the association between Hsf1 and the mediator complex

Since phosphorylation does not disrupt the interaction with Hsp70, we suspected that it serves to increase the ability of Hsf1 to directly activate transcription. To test this, we tagged full-length wild type Hsf1, Hsf1$^{\Delta po4}$ and Hsf1$^{PO4*}$ with mKate2 and monitored their ability to activate the HSE-YFP reporter when expressed as the only versions of Hsf1 in the cell as a function of estradiol in the absence of heat shock (**Figure 5D**). In contrast to the indistinguishable transcriptional responses of the decoys, these three versions of full length Hsf1 displayed distinct induction curves as a function of their absolute expression level (**Figure 5E**). Hsf1$^{\Delta po4}$ was a slightly weaker activator than wild type Hsf1, while Hsf1$^{PO4*}$ was by far the strongest activator with the steepest slope (**Figure 5E**).

We hypothesized that Hsf1$^{PO4*}$ exerts its increased activity by more efficiently recruiting the transcriptional machinery. Since Hsf1 is known to engage RNA polymerase II via interaction with the Mediator complex (**Kim and Gross, 2013**), we performed ChIP-seq of the Mediator subunit Med4 in wild type, Hsf1$^{\Delta po4}$ and Hsf1$^{PO4*}$ cells. Indeed, recruitment of Med4 to the promoters of HDGs, such as *HSP82*, *SSA4* and *SSA1*, was very clearly a function of phosphorylation state, with Hsf1$^{PO4*}$ recruiting the most Med4, wild type Hsf1 recruiting less, and finally Hsf1$^{\Delta PO4}$ showing very little recruitment (**Figure 5F**, **Figure 5—figure supplement 1**).

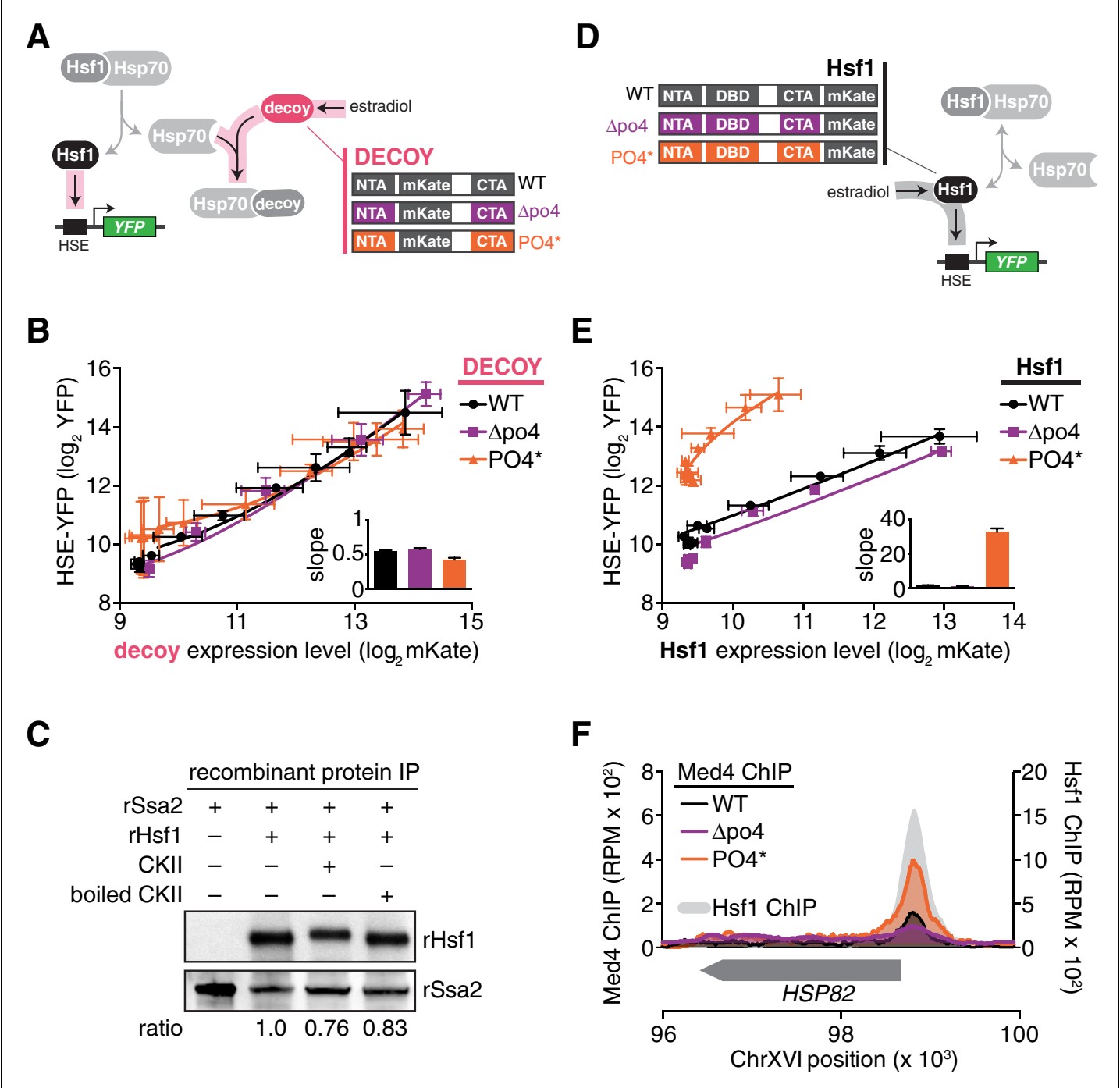

**Figure 5.** Hsp70 binding and phosphorylation are uncoupled Hsf1 regulatory mechanisms. (A) Schematic cartoon of decoy constructs based on wild type Hsf1 (WT, black), Hsf1$^{\Delta po4}$ (Δpo4, purple) and Hsf1$^{PO4*}$ (PO4*, orange). The various decoys will activate endogenous Hsf1 in proportion to their affinity for Hsp70. (B) Measurement of the HSE-YFP reporter by flow cytometry in cells expressing decoy constructs derived from wild type Hsf1, Hsf1$^{\Delta po4}$ or Hsf1$^{PO4*}$ as a function of the expression level of each decoy (mKate fluorescence). Data points are the average of median YFP and mKate values for three biological replicates, and error bars are the standard deviation. See Materials and methods for assay and analysis details. The slope of the input-output curves are plotted (inset). (C) IPs of recombinant proteins were performed with 3xFLAG-rSsa2 as bait and analyzed by Western blot. rHsf1 was pre-incubated with ATP alone or in the presence of ATP and either active casein kinase II (CKII) or boiled CKII. Blots were probed with an anti-FLAG antibody to recognize recombinant rSsa2 (top) and with an anti-HIS antibody to recognize recombinant rHsf1 (bottom). The numbers below the blots indicate the normalized ratio of Hsf1/Ssa2. (D) Schematic cartoon of full-length overexpression constructs for wild type Hsf1 (WT, black), Hsf1$^{\Delta po4}$ (Δpo4, purple) and Hsf1$^{PO4*}$ (PO4*, orange), each with mKate2 fused to its C-terminus. The full-length constructs will activate the HSE-YFP reporter in proportion to their transcriptional activity. (E) Measurement of the HSE-YFP reporter by flow cytometry in cells expressing full length

*Figure 5 continued on next page*

*Figure 5 continued*

constructs of wild type Hsf1, Hsf1$^{\Delta po4}$ or Hsf1$^{PO4*}$ tagged at their C-termini with mKate2 as a function of expression level as in **B**. (**F**) ChIP-seq for Med4-3xFLAG-V5, a component of the Mediator complex, in basal conditions in wild type Hsf1, Hsf1$^{\Delta po4}$, and Hsf1$^{PO4*}$ cells at the *HSP82* locus. Wild type Hsf1-3xFLAG-V5 ChIP-seq was also performed in basal conditions (gray filled curve). See *Figure 5—figure supplement 1* for more loci.
The following figure supplement is available for figure 5:

**Figure supplement 1.** Hsf1$^{PO4*}$ recruits Mediator more efficiently than wild type Hsf1 or Hsf1$^{\Delta po4}$.

## Phosphorylation accounts for sustained Hsf1 activity during heat shock

The mathematical model, which ignored phosphorylation, failed to account for the persistent Hsf1 activity that followed Hsp70 re-association during the heat shock time course (*Figure 1D*). To determine if Hsf1 phosphorylation could explain this sustained activity, we monitored Hsf1 phosphorylation kinetics via mobility shift of Hsf1-3xFLAG-V5 throughout a heat shock time course by Western blot. Rather than coinciding with the dissociation of Hsp70, which occurs within the first five minutes following temperature upshift (*Figure 1A*), Hsf1 phosphorylation peaked after 20 min and was maintained out to at least two hours (*Figure 6A*). Thus, the timing of Hsf1 phosphorylation matches the second phase of Hsf1 transcriptional activity (*Figure 6B*). This result suggests that Hsf1 phosphorylation can drive increased transcription even when Hsf1 is bound to Hsp70. Consistent with phosphorylation driving the second phase of Hsf1 transcriptional activity, Hsf1$^{\Delta po4}$ completely lacked sustained activity at the later time points (*Figure 6B*). Incorporating phosphorylation into the mathematical model as a 'positive gain' according to the experimentally determined kinetics (*Figure 6—figure supplement 1A*, see Materials and methods) allowed the model to recapitulate the activation dynamics of both wild type Hsf1 and Hsf1$^{\Delta po4}$ throughout the heat shock time course (*Figure 6—figure supplement 1B*). Taken together, the data presented here support a model in which Hsf1 integrates negative feedback regulation by Hsp70 and positive fine-tuning by phosphorylation to dynamically control the heat shock response (*Figure 6D*).

## Discussion

It has been suggested that the heat shock response operates as a feedback loop, in which Hsf1 activity is determined by the abundance of free chaperones (*Voellmy and Boellmann, 2007*). However, the elegance of this model has perhaps overshadowed the lack of data to support it. Here we provide multiple lines of evidence for a chaperone titration model in budding yeast. Specifically, we showed that the Hsp70 chaperone binds to Hsf1 in basal conditions, dissociates during the acute phase of heat shock, and subsequently re-associates at later time points, thus providing the first direct evidence of a dynamic Hsf1 'switch' (*Figure 1A*). Furthermore, we reconstituted the interaction between Hsp70 and Hsf1 in vitro with recombinant proteins and partially disrupted the complex with a hydrophobic peptide (*Figure 1A*). We then constructed a minimal mathematical model of the Hsp70 feedback loop, and showed that it recapitulated the dynamics of Hsf1 transcriptional activity during heat shock (*Figure 1B–D*). The model also correctly predicted that synthetic perturbations to the feedback loop, such as adding a 'decoy' of Hsf1, would activate the endogenous Hsf1 response via Hsp70 titration (*Figure 2*). Finally, we provided independent genetic support for the model by showing that increased expression of Hsp70 and Hsp40 (an Hsp70 co-factor) suppresses the growth impairment caused by Hsf1 overexpression (*Figure 3*). Thus, biochemical, genetic and computational approaches converged to support a model in which Hsp70 and Hsf1 form a feedback loop that controls heat shock response activation.

While our results are consistent with studies reporting biochemical and genetic interactions between Hsf1 and Hsp70 (*Abravaya et al., 1992*; *Baler et al., 1992*, *1996*; *Brandman et al., 2012*; *Guisbert et al., 2013*; *Ohama et al., 2016*; *Shi et al., 1998*), they are inconsistent with other reports that implicate Hsp90 as a major repressor of Hsf1 activity (*Brandman et al., 2012*; *Duina et al., 1998*; *Guo et al., 2001*; *Zou et al., 1998*). Biochemically, our inability to detect Hsp90 binding to Hsf1 could be due to the serial affinity purification strategy that we employed: if Hsp90 weakly associates with Hsf1, we would likely lose the interaction during the two-step purification. However, the

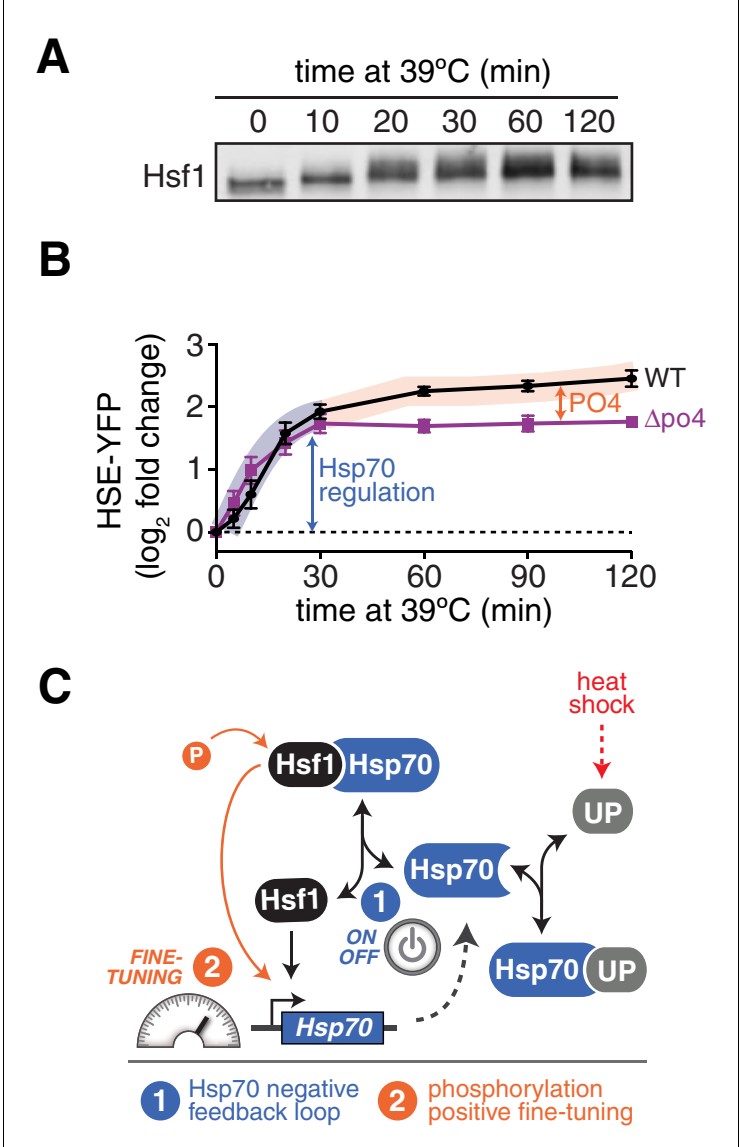

**Figure 6.** Hsf1 phosphorylation accounts for sustained Hsf1 transcriptional activity during heat shock. (**A**) Western blot of Hsf1 phosphorylation, indicated by its mobility shift, over time following temperature upshift. (**B**) The HSE-YFP reporter was measured by flow cytometry in wild type Hsf1 and Hsf1$^{\Delta po4}$ cells over time following upshift to 39°C. Data points are the average of median YFP values for three biological replicates, and error bars are the standard deviation. See Materials and methods for assay and analysis details. The sustained activation attributable to phosphorylation is depicted as the orange segment of the wild type curve. (**C**) Cartoon schematic of the integrated phosphorylation/chaperone titration model of Hsf1 regulation. Phosphorylation (PO4) increases the transcriptional activity of Hsf1 in a manner uncoupled from the Hsp70 feedback loop.

The following figure supplement is available for figure 6:

**Figure supplement 1.** Inclusion of the role of phosphorylation in the mathematical model of Hsf1 regulation.

combination of the lack of biochemical evidence with the genetic result that Hsp90 overexpression is unable to suppress Hsf1 overexpression suggests a parsimonious explanation that Hsp90 simply does not repress yeast Hsf1. Using our Hsp70-centric model, we can explain the genetic studies in which loss of Hsp90 function activates Hsf1 (*Brandman et al., 2012*; *Duina et al., 1998*) by

supposing that reduced Hsp90 leads to increased levels of unfolded proteins that titrate Hsp70 away from Hsf1.

Phosphorylation of Hsf1, which is a hallmark of the heat shock response, is a second longstanding mechanism proposed to regulate Hsf1 (*Sorger and Pelham, 1988*). We identified 73 sites of phosphorylation on Hsf1 across various conditions (*Figure 4—source data 1*). Remarkably, however, Hsf1 retained its essential basal functionality and heat shock-induced activity in the complete absence of phosphorylation (*Figure 4*). Consistent with this observation, human Hsf1 also remained heat shock-inducible following mutation of a subset of its phosphorylation sites (*Budzynski et al., 2015*). Despite its qualitative functionality in the absence of phosphorylation, Hsf1$^{\Delta po4}$ was quantitatively impaired in its ability to induce its target genes during heat shock (*Figure 4E*). Conversely, mimicking hyper-phosphorylation increased basal expression of the Hsf1 target regulon without disrupting its heat shock inducibility (*Figure 4F,G*). Moreover, the number of phospho-mimetic residues correlated with transcriptional output (*Figure 4I*). Thus, in contrast to prevailing models suggesting that phosphorylation is required for activation, we conclude that phosphorylation is not the switch that activates Hsf1; rather, phosphorylation is a positive fine-tuner that amplifies the transcriptional activity of Hsf1. We propose that increased negative charge in the transcriptional activation domains of Hsf1, endowed by phosphorylation or phospho-mimetic mutations, increases the ability of Hsf1 to recruit the Mediator complex and initiate transcription (*Figure 5F*). In this manner, phosphorylation renders the activation domains into the 'acid blobs' that have long been associated with potent transcriptional activators (*Sigler, 1988*).

Although this work yields a coherent synthesized model for Hsf1 regulation during heat shock, a number of interesting questions remain. These include identifying the molecular determinants of the Hsf1•Hsp70 interaction, defining the mechanism by which Hsp70 represses Hsf1 under basal conditions and whether this mechanism also applies to Hsf1 deactivation, determining potential roles for other chaperones (such as Hsp40, Hsp90 and chaperonins) in Hsf1 regulation – particularly in different evolutionary lineages – and identifying the kinases that phosphorylate Hsf1. On the latter, many kinases have been shown to regulate Hsf1 in various conditions, but the pathways that converge on Hsf1 have yet to be systematically defined. Given the low level of conservation of the Hsf1 activation domains (*Anckar and Sistonen, 2011*) and the preponderance of S/T residues in putatively unstructured, solvent exposed regions of the protein, many kinases could potentially find a substrate site on Hsf1. Defining the cohort of active kinases present in the nucleus in various conditions would provide a useful starting point to identify the kinases responsible for phosphorylating Hsf1.

Although both Hsp70 binding and phosphorylation contribute to regulating Hsf1 activity, they are independent events that exert orthogonal control. Hsf1 phosphorylation does not interfere with Hsp70 binding (*Figure 5B,C*), and the kinetics of Hsf1 phosphorylation are delayed with respect to Hsp70 dissociation, with phosphorylation peaking after Hsp70 has re-associated with Hsf1 (*Figures 1A* and *6A*). These two distinct regulatory modes correlate with an immediate surge in Hsf1 transcriptional activity followed by a sustained moderate level of activity (*Figure 6B*). Uncoupled regulation by chaperone binding and phosphorylation allows Hsf1 to function as an integration hub. Hsf1 can directly link to the proteostasis network by sensing the availability of Hsp70 as a proxy for protein folding conditions as well as respond to signals from multiple kinase pathways that convey information about other intracellular and extracellular conditions, such as oxidative stress and nutrient availability (*Hahn and Thiele, 2004*; *Yamamoto et al., 2007*). In this manner, proteotoxic stress could activate Hsf1 without phosphorylation, and kinases could activate Hsf1 without chaperone dissociation.

In this study, we employed a suite of approaches that allowed us to converge on a simple model of Hsf1 regulation and the ensuing dynamics of the heat shock response in budding yeast. However, given the poor conservation of the regulatory domains of Hsf1, combined with the promiscuity of chaperone protein interactions and the ease with which phosphorylation sites are gained and lost through evolution, Hsf1 regulation has the potential to be rewired in different organisms and perhaps even in different cell types within the same organism (*Guisbert et al., 2013*). Nevertheless, this work for the first time defines a regulatory scheme that synthesizes the roles of both canonical Hsf1 regulatory mechanisms, and as such can serve as both a precedent and template for the dissection of Hsf1 regulation in other cellular models. With a quantitative understanding of how Hsf1 is regulated when it is functioning properly, we can begin to unravel how it breaks down in neurodegenerative disorders and is usurped in cancer.

## Materials and methods

### Yeast strains and plasmids

Yeast strains and plasmids used in this work are described in *Supplementary files 1* and *2*, respectively. All strains are in the W303 genetic background. PCR-mediated gene deletion and gene tagging was carried out as described (*Longtine et al., 1998*).

### Serial anti-FLAG/anti-V5 immunoprecipitation

The serial immunoprecipitation procedure is described in more detail at Bio-protocol (*Zheng and Pincus, 2017*). 250 ml of cells were grown to $OD_{600}$ = 0.8 in YPD media at 25°C with shaking. Basal condition samples were collected by filtration and filters were snap frozen in liquid $N_2$ and stored at −80°C. For heat shocked samples, 250 ml of YPD pre-warmed to 53°C was added to the 250 ml culture to immediately raise the temperature to 39°C and cultures were incubated with shaking for the indicated times (5, 15, 30 or 60 min) before being collected as above. Cells were lysed frozen on the filters in a coffee grinder with dry ice. After the dry ice was evaporated, lysate was resuspended in 1 ml IP buffer (50 mM Hepes pH 7.5, 140 mM NaCl, 1 mM EDTA, 1% triton x-100, 0.1% DOC, complete protease inhibitors), transferred to a 1.5 ml tube and spun to remove cell debris. Clarified lysate was transferred to a fresh tube and serial IP was performed. First, 50 µl of anti-FLAG magnetic beads (50% slurry, Sigma) were added, and the mixture was incubated for 2 hr at 4°C on a rotator. Beads were separated with a magnet and the supernatant was removed. Beads were washed three times with 1 ml IP buffer and bound material eluted with 1 ml of 1 mg/ml 3xFLAG peptide (Sigma-Aldrich, St. Louis, MO) in IP buffer by incubating at room temperature for 10 min. Beads were separated with a magnet and eluate was transferred to a fresh tube. Next, 25 µl of anti-V5 magnetic beads (50% slurry, MBL International) were added and the mixture was incubated for 2 hr at 4°C on a rotator. Beads were separated with a magnet and the supernatant was removed. Beads were washed five times with 1 ml IP buffer. Bound material was eluted by adding 75 µl of SDS-PAGE sample buffer and incubating at 95°C for 10 min. Beads were separated with a magnet and sample was transferred to a fresh tube for analysis. Control samples included an untagged strain and a strain expressing GFP-3xFLAG-V5.

### Mass spectrometry

Mass spectrometry analysis was performed at the Whitehead Proteomics Core Facility. IP eluates were digested with trypsin and analyzed by liquid chromatography (NanoAcuity UPLC) followed by tandem mass spectrometry (Thermo Fisher LTQ). Mass spectra were extracted and analyzed by MASCOT searching the yeast proteome modified to include the 3xFLAG-V5 tagged bait proteins with a fragment ion mass tolerance of 0.8 Da and a parent ion tolerance of 20 PPM. Proteins were identified by Scaffold v4.4.1 to validate peptide and protein IDs. Peptides were accepted with a confidence of >95% and protein IDs were accepted only if they could be established at99% confidence and contained at least two peptides. Proteins that contain indistinguishable peptides were clustered. Quantification was performed for Hsf1 and Ssa1/2 using the 'top three' peptide total ion current method (*Grossmann et al., 2010*).

### Western blotting

15 µl of each IP sample was loaded into 4–15% gradient SDS-PAGE gels (Bio-Rad). The gels were run at 25 mA for 2 hr, and blotted to PVDF membrane. After 1 hr blocking in Li-Cor blocking buffer, the membrane was incubated with anti-FLAG primary antibody (SIGMA, F3165) for 1 hr, anti-Ssa1/2 (gift from V. Denic) or anti-HIS antibody (all 1:1000 dilutions). The membranes were washed three times with TBST. The proteins were probed by anti-mouse-800 IgG (Li-Cor, 926–32352, 1:10000 dilution). The fluorescent signal scanned with the Li-Cor/Odyssey system. For the heat shock time course, cells expressing Hsf1-3xFLAG-V5 were grown to $OD_{600}$0.8 in 25 ml YPD at 25°C. At time t = 0, 25 ml of 53°C media was added to instantly bring the culture to 39°C and then the culture was incubated with shaking at 39°C. 5 ml samples were collected at each time point by centrifugation. Pellets were boiled with 2X SDS loading buffer for 10 min. Total protein concentration was measured by NanoDrop and an equal amount of each sample was loaded into 7.5% SDS-PAGE gel and otherwise processed as above.

## Recombinant protein expression and purification

Full-length wild type *HSF1* was cloned into pET32b with a C-terminal 6x-HIS tag and sequenced. Site-directed mutagenesis was performed to introduce the S225A and S225D mutations. Full length *SSA2* was cloned into pET32b with an N-terminal 6xHIS-3xFLAG tag. The plasmids were transformed into BL21(DE3) cells (Invitrogen). One liter of cells at OD600 = 0.4 were induced with 1 mM IPTG for 3 at 37°C. Cells were lysed by sonication, protein was purified with Ni-NTA agarose (Qiagen) and eluted with imidazole.

## In vitro immunoprecipitation and competition with Aβ42

6xHIS-3xFLAG-Ssa2 and Hsf1-6xHIS were mixed with each at a final concentration of 5 µM in 100 µl of IP buffer alone, in the presence of 1 mM ATP or in the presence of 25 µM Aβ42. Reactions were incubated at room temperature for 10 min. 25 µl of anti-FLAG magnetic beads were added and incubated for 15 min before magnetic separation. The unbound fraction was removed, beads were washed three times with 1 ml IP buffer and proteins were eluted by incubating at 95°C in SDS-PAGE sample buffer.

## Mathematical modeling

### Overview and assumptions

We developed a minimal kinetic model for an Hsp70-mediated feedback loop, consisting of four protein species: Hsf1, Hsp70, unfolded protein (UP), and reporter protein YFP (*Figure 1—figure supplement 2*). According to the model, free Hsp70 binds to Hsf1 in a transcriptionally inactive complex, while free Hsf1 induces production of Hsp70 (and the YFP reporter). In addition to binding to Hsf1, Hsp70 can also bind to unfolded protein (UP) and become titrated away from Hsf1. There are a few key assumptions to the model. First, the model assumes Hsp70 cannot bind to both Hsf1 and UP simultaneously. Second, the dilution rate of YFP molecules due to cell division is assumed to be negligible ($k_{dil} \sim 0$); this is because the time course for our simulations and reporter heat shock assays is short (~1 cell generation) and YFP molecules are relatively long lived (*Figure 1D*). Third, this initial implementation of the model ignores any potential role for Hsf1 phosphorylation. Fourth, for simplicity, the model assumes the on-rate constants of Hsp70 for both clients (UP and Hsf1) are the same (*Mayer and Bukau, 2005*; *Pierpaoli et al., 1998*). Finally, binding of Hsf1 to DNA driving transcription of *Hsp70* or *YFP* is modeled by a Hill equation with a Hill coefficient of $n = 3$ as a simple representation of Hsf1 trimerization (*Sorger and Nelson, 1989*).

### Differential equations

Our model consisted of a system of six coupled ordinary differential equations:

$$\frac{d[HSP]}{dt} = k_2[HSP \cdot Hsf1] - k_1[HSP][Hsf1] + (k_4 + k_5)[HSP \cdot UP] - k_3[HSP][UP] + \beta \frac{[Hsf1]^n}{K_d^n + [Hsf1]^n}$$

$$\frac{d[Hsf1]}{dt} = k_2[HSP \cdot Hsf1] - k_1[HSP][Hsf1]$$

$$\frac{d[UP]}{dt} = k_4[HSP \cdot UP] - k_3[HSP][UP]$$

$$\frac{d[HSP \cdot HSF1]}{dt} = k_1[HSP][Hsf1] - k_2[HSP \cdot Hsf1]$$

$$\frac{d[HSP \cdot UP]}{dt} = k_3[HSP][UP] - (k_4 + k_5)[HSP \cdot UP]$$

$$\frac{d[YFP]}{dt} = \beta \frac{[Hsf1]^n}{k_d^n + [Hsf1]^n} - k_{dil}[YFP]$$

with rate constants $k_1$-$k_5$, transcriptional activation rate $\beta$, and Hsf1-DNA affinity $K_d$.

## Parameter assignments and initial conditions

To assign these parameter values, we used our IP/MS Hsp70 dissociation data as a constraint (*Figure 1C*). Specifically, we chose a range of values for each parameter (Figure S2C). We then ran simulations for all possible combinations of parameters equally distributed across the ranges ($7 \times 10^5$ total parameters) to select parameter sets that satisfy three features of our Hsp70 dissociation data. First, Hsf1 must be bound to Hsp70 in basal conditions at 25°C. Second, the Hsf1•Hsp70 complex must dissociate to ≤10% of its basal level within five minutes following a shift from 25°C to 39°C. Third, the Hsf1•Hsp70 complex must re-associate to ≥90% of its initial level within 60 min following a shift from 25°C to 39°C. As an initial condition, we set the ratio of the level of Hsp70:Hsf1 to 500:1, which is in the range of measured values (*Chong et al., 2015*; *Kulak et al., 2014*). We obtained parameter sets that satisfied these constraints (Figure S2C), from which we selected final parameters that quantitatively recapitulated the transient dissociation behavior of Hsp70 (*Table 1*). Satisfyingly, this analysis revealed that most parameters tolerated a broad range of values (Figure S2C), provided that Hsf1 induces transcription cooperatively (i.e. one of our assumptions), which is consistent with prior work showing that Hsf1 binds DNA as a trimer (*Sorger and Nelson, 1989*).

To simulate responses for different heat shock temperatures, we used a simple function to relate the concentration of UPs with temperature (Figure S2B). According to calorimetry and other previous work investigating cellular protein unfolding during heat shock, the relationship between UP and temperature can be simply described by an exponential function for low temperatures (25°C–50°C) (*Lepock et al., 1993*; *Scheff et al., 2015*). Specifically, we used an exponential function that passes through our fixed value of [UP] at 39°C (the value used for the parameter screen described above, *Table 2*), a small basal value of [UP] at 25°C ([UP]$_o$ = 0.52), and [UP] values at other temperatures that generally capture steady state YFP reporter outputs. [UP] for other temperatures could then be obtained using this function, and used as initial values for simulating upshifts to other temperatures, such 35°C and 43°C (*Figure 1D*).

## Simulation details

All simulations were run on MATLAB; ode23s was used to solve our coupled ordinary differential equations for a length of two hours. Code is provided as a supplementary .zip file containing 7 .m files.

## Addition of the Hsf1 decoy

To simulate an Hsf1 decoy, we added two additional species: Hsf1_dec (free) and Hsf1_dec•Hsp70 (bound to Hsp70). The model assumes the decoy has the same binding kinetics for Hsp70 as that of Hsf1, but that the decoy does not bind DNA and drive gene expression. To simulate a titration (or estradiol induction) of either WT Hsf1 or the decoy (*Figure 2C,D*), we varied the initial concentration of these species across a range of concentrations and ran simulations for each.

**Table 1.** Model parameter values.

| Parameter | Value | Description |
| --- | --- | --- |
| $k_1$, $k_3$ | 166.8 min$^{-1}$ a.u.$^{-1}$ | Client•Hsp70 on-rate constant |
| $k_2$ | 2.783 min$^{-1}$ | Hsp70•HSF1 off-rate constant |
| $k_4$ | 0.0464 min$^{-1}$ | Hsp70•UP off-rate constant |
| $k_5$ | 4.64e-7 min$^{-1}$ | Degradation rate of UP by Hsp70•UP |
| $\beta$ | 1.778 min$^{-1}$ | Transcriptional activation rate |
| $K_d$ | 0.0022 a.u. | Dissociation constant of Hsf1-DNA interaction |
| $k_{dil}$ (fixed) | 0 min$^{-1}$ | Dilution rate of YFP |
| n (fixed) | 3 | Hill coefficient |

**Table 2.** Model initial conditions.

| Species | Initial value (a.u.) | Description |
|---|---|---|
| $[HSP]_0$ | 1 | Free Hps70 |
| $[Hsf1]_0$ | 0 | Free Hsf1 |
| $[HSP \bullet Hsf1]_0$ | 1/500 | HSP70•Hsf1 complex |
| $[HSP \bullet UP]_0$ | 0 | Hsp70•UP complex |
| $[YFP]_0$ | 3 | Initial YFP concentration |
| $[UP]_0$ (@ 39°C) | 10.51 | UP concentration at 39°C |

## Inclusion of the role of Hsf1 phosphorylation

Following temperature upshift, Hsf1 is fully phosphorylated within approximately one hour (*Figure 6A*), and phosphorylated Hsf1 leads to increased recruitment of transcriptional machinery (*Figure 5*). To model this, we assumed that the transcriptional activation rate, $\beta$, correlates with phosphorylation state, and allowed $\beta$ to vary over time in accordance with Hsf1 phosphorylation (*Figure 6A*). Specifically, we used a sigmoid function to evolve $\beta$ from an initial value (pre-phosphorylated) to a final value (phosphorylated), and applied this function to our heat shock simulations (Figure S2D,E).

## HSE-YFP reporter heat shock assays

All heat shock reporter assays were performed with untagged Hsf1 and mutants. For time course reporter inductions, 500 μl of $OD_{600}$ = 0.1 cells were incubated at 39 with shaking on a thermo-mixer in 1.5 ml tubes. At designated time points, 50 μl samples were taken and cycloheximide was added at 50 μg/ml to arrest translation. Arrested cells were incubated at 30°C for 2 hr to allow fluorophores to mature. Samples were measured by flow cytometry, and population medians were computed with FlowJo. Each data point is the mean of three or four biological replicates. Error bars are the standard deviation.

For basal versus heat shock experiments, we developed a protocol in which samples were pulsed with repeated 15 min heat shocks at 39°C followed by recovery at 25 for 45 min. As a control, a sample was kept at 25°C during the same time as heat-shocked pulses experiment. All experiments were performed using C1000 Touch Thermal Cycler (Bio-Rad). Cells had been serially diluted five times (1:5) in SDC and grown overnight at room temperature. Cells in logarithmic phase were chosen the next morning for the experiment and 50 μl of each strain was transferred to two sets of PCR tubes and thermal cycled as described above. After that samples were transferred to 96-well plates with 150 μl of 1xPBS. HSE activity was measured using flow cytometry (BD LSRFortessa) and data analyzed using Flowjo as above.

## Estradiol dose response assays

Cells bearing the HSE-YFP reporter and a chimeric transcription factor, GEM, consisting of the Gal4 DNA binding domain, the human estrogen receptor and the Msn2 activation domain (*Pincus et al., 2014*) were transformed with either the Hsf1 decoy, phospho-mutant decoys or mKate alone expressed from the *GAL1* promoter and integrated as single copies in the genome. Full-length wild type Hsf1 and full-length C-terminally mKate tagged wild type Hsf1 and phospho-mutant constructs were also expressed under the control of the *GAL1* promoter and integrated as single copies into the genome, but in a strain background that additionally contained a genomic deletion of *HSF1* and a *CEN/ARS URA3*-marked plasmid bearing *HSF1*. Upon integration of these constructs, the plasmid was counter-selected on 5-FOA. The GEM construct makes the *GAL1* promoter 'leaky' enough that the cells are viable in the absence of estradiol and the presence of glucose. Cells were first grown to saturation overnight in synthetic media with dextrose and complete amino acids (SDC). To assay for growth impairment and transcriptional activity as a function of expression level, cells in 10 different concentrations of estradiol ranging from 512 nM to 1 nM in SDC across a two-fold serial dilution series in deep well 96 well plates (5 μl of saturated culture diluted into 1 ml of each estradiol

concentration). Each dose of estradiol was performed in triplicate. To prevent saturation of the cultures, each estradiol concentration was serially diluted 1:4 into media with the same concentration of estradiol. Following 18 hr of growth, cell counts, HSE-YFP levels and mKate levels were measured by flow cytometry by sampling 10 μl of each culture (BD LSRFortessa equipped with a 96-well plate high-throughput sampler) and the data were analyzed in FlowJo. Relative growth rates were calculated by dividing the number of cells in each concentration of estradiol by the 1 nM estradiol counts.

## Ubc9ts aggregation reporter and live cell microscopy

Cells expressing YFP-Ubc9ts and the Hsf1 decoy were grown overnight in complete media with raffinose. Cells were induced with 2% galactose for 4 hr to induce expression of YFP-Ubc9ts and the decoy. One sample was untreated and the other was heat shocked for 15 at 39°C. 96 well glass bottom plates were coated with 100 μg/ml concanavalin A in water for 1 hr, washed three times with water and dried at room temperature. 80 μl of low-density cells were added to a coated well. Cells were allowed to settle and attach for 15 min, and unattached cells were removed and replaced with 80 μl SD media. Imaging was performed at the W.M Keck Microscopy Facility at the Whitehead Institute using a Nikon Ti microscope equipped with a 100×, 1.49 NA objective lens, an Andor Revolution spinning disc confocal setup and an Andor EMCCD camera.

## Dilution series spot growth assays

Yeast strains containing wild type or mutated versions of *HSF1* as the only copy of the gene in the genome were grown overnight in YPD. They were diluted to an identical final $OD_{600} = 0.3$ in phosphate buffered saline (1xPBS) and serially diluted 1:5 in 1xPBS. 3.5 μl of each diluted yeast culture was spotted on the appropriate plate. Photographs were taken after two days of growth at 30 or 37°C.

## Cell cycle stage analysis by tubulin immunofluorescence

Tubulin immunofluorescence was performed in the presence or absence of GAL-overexpressed Hsf1 following release from alpha factor arrest as described (*Kilmartin and Adams, 1984*).

## DNA content analysis

DNA content analysis was performed in the presence or absence of GAL-overexpressed Hsf1 following release from alpha factor arrest as described (*Hochwagen et al., 2005*).

## Hsf1 phosphorylation site identification

Hsf1-3xFLAG-V5 was immunoprecipitated as described above following the appropriate treatment through the first (anti-FLAG) purification step. Rather than eluting with 3xFLAG peptide, samples were incubated at 95°C in SDS-PAGE sample buffer to elute Hsf1. The samples were run on SDS-PAGE and stained with coomassie. All phosphorylation site identification was outsourced to Applied Biomics (Hayward, CA). We sent them samples (cut bands from coomassie-stained gels) and received excel sheets with phospho-peptides identified, called sites, coverage stats, and neutral loss spectra.

## Site-directed mutagenesis

Site-directed mutagenesis was performed with QuickChange according to the manufacturer's directions (Agilent).

## Synthetic genes and Gibson assembly

*En masse* mutational analysis was possible because of gene synthesis. We ordered gBlocks from IDT containing regions of Hsf1 with all S/T codons mutated to alanine. The C-terminal portion required codon optimization in order to remove repetitive sequence to allow synthesis. Originally, restriction sites were introduced at the boundaries of the regions to enable cut-and-paste combinatorial cloning. Finally, all restriction sites were removed by assembling the fragments via Gibson assembly. Gibson assembly was performed as directed by the manufacturer (NEB).

## $^{32}$P incorporation

Strains bearing Hsf1-3xFLAG-V5 or Hsf1$^{\Delta po4}$-3xFLAG-V5 expressed under estradiol control were grown in YPD liquid media to OD$_{600}$ = 0.5. Then protein expression was induced with 1 μM estradiol for 2. Cells were pelleted and washed with 50 ml SDC media without phosphate. Cells were finally resuspended in 15 ml SDC media without phosphate, and incubated at room temperature for 30 min. 50 μCi of $^{32}$P-orthophosphate was added into each culture and the cells were incubated for 15 min. The samples were heat shocked at 39°C for 30 min, harvested, and Hsf1 was IP'ed as above. All the protein was loaded into an SDS-PAGE gel. After blotting the proteins to the PVDF membrane, the signal was detected by FujiFilm BAS-2500 system. The same membranes were then blotted for total Hsf1 as described above.

## mRNA deep sequencing (RNA-seq)

5 ml of cells were grown to OD$_{600}$ = 0.5 and treated with the designated condition. Cells were spun and pellets were snap frozen and stored at −80°C. Pellets were thawed on ice, and total RNA was purified via phenol/chloroform separation using phase lock tubes (five prime) followed by ethanol precipitation (Pincus et al., 2010). Total RNA samples were submitted to the Whitehead Genome Technology Core where polyA + RNA was purified, fragmented and sequencing libraries were prepared with barcoding. 12 samples were multiplexed in each lane of an Illumina Hi-Seq 2500 and deep sequencing was performed. Reads were assigned by the barcode to the appropriate sample. Data was processed using a local version of the Galaxy suite of next-generation sequencing tools. Reads were groomed and aligned to the *S. cerevisiae* orf_coding reference genome (Feb. 2011) using Tophat, transcripts were assembled and quantified using Cufflinks and fold changes were computed using Cuffdiff (Trapnell et al., 2012).

## Electrophoretic mobility shift assay

An oligo containing four repeats of the HSE was synthesized with and without a 3' fluorescent probe (IRDye800) by IDT. The reverse complement was also synthesized and the oligos were annealed by heating to 95 followed by room temperature cooling. Labeled dsDNA was prepared at 5 μM and unlabeled at 50 μM. 1.5 μg of each Hsf1 prep was added to 1 μl of labeled DNA or 1 μl of labeled DNA plus 1 μl of unlabeled DNA in 10 μl total volume. Reactions were incubated at room temperature for 5 min. 2 μl 6x DNA loading dye were added and samples were loaded into 4–20% TBE gels (Bio Rad). Gels were run at 30 mA for 1 hr and scanned on the LiCor. Images were analyzed and % shifted oligo was quantified in ImageJ

## In vitro phosphorylation

Casein kinase II (CKII) was used to phosphorylate rHsf1 in vitro as described by the manufacturer (NEB). As a control CKII was boiled to denature and deactivate it prior to incubation with rHsf1.

## Chromatin immunoprecipitation and deep sequencing (ChIP-seq)

50 ml of cells were fixed with addition of 1% formaldehyde for 20 min at room temperature followed by quenching with 125 mM glycine for 10 min. Cells were pelleted and frozen in liquid N$_2$ and stored at −80°C. Cells were lysed frozen in a coffee grinder with dry ice. After the dry ice was sublimated, lysate was resuspended in 2 ml ChIP buffer (50 mM Hepes pH 7.5, 140 mM NaCl, 1 mM EDTA, 1% triton x–100, 0.1% DOC) and sonicated on ice 10 times using a probe sonicator (18W, 30 s on, one minute off). 1 ml was transferred to a 1.5 ml tube and spun to remove cell debris. Input was set aside, and a serial IP was performed. First, 25 μl of anti-FLAG magnetic beads (50% slurry, Sigma) were added the mixture was incubated for 2 hr at 4°C on a rotator. Beads were separated with a magnet and the supernatant was removed. Beads were washed five times with 1 ml ChIP buffer (5 min incubations at 4°C between each wash) and bound material eluted with 1 ml of 1 mg/ml 3xFLAG peptide (Sigma) in ChIP buffer by incubating at room temperature for 10 min. Beads were separated with a magnet and eluate was transferred to a fresh tube. Next, 25 μl of anti-V5 magnetic beads (50% slurry, MBL International) were added and the mixture was incubated for 2 hr at 4°C on a rotator. Beads were separated with a magnet and the supernatant was removed. Beads were washed three times with ChIP buffer, followed by a high salt wash (ChIP buffer +500 mM NaCl) and a final wash in TE. Bound material was eluted with 250 μl TE +1% SDS by incubating at 65°C for 15 min.

Beads were separated with a magnet and eluate was transferred to a fresh tube and incubated overnight at 65°C to reverse crosslinks. Protein was degraded by adding 250 µl 40 µg/ml proteinase K in TE (supplemented with GlycoBlue) and incubating at 37°C for 2 hr. DNA fragments were separated from protein by adding 500 µl phenol/chloroform/isoamyl alcohol (25:24:1), and the aqueous layer was added to a fresh tube. 55 µl of 4M LiCl was added along with 1 ml of 100% EtOH, and DNA was precipitated at −80°C overnight. DNA was pelleted by spinning for 30 min at 4°C and resuspended in 50 µl TE. Sequencing libraries were prepared by the WIGTC, and sequenced on the Illumina Hi-Seq 2500.

## Acknowledgements

We thank E Spooner and the Whitehead Institute Mass Spectrometry facility; E Solís for suggesting that S225 may be the only essential serine in Hsf1; V Denic for the gift of the Ssa1/2 antisera and critical reading of the manuscript; H Hadjivassiliou for help with PyMol, beneficial discussions and critical reading of the manuscript; R Park for help establishing the gel-shift assay; J Falk and A Amon for assistance with cell cycle experiments; G Fink, H Lodish and AJaeger for critical reading of the manuscript; J Pandey, S Lourido, G Victora and members of the Lourido and Victora labs for valuable discussions; S Lindquist, and members of the her lab for helpful comments and suggestions. We are grateful to T Volkert and the Whitehead Genome Technology Core; P Wisniewski and the Whitehead Flow Cytometry Facility; W Salmon and the Keck Microscopy Facility; I Barassa, P Thiru and G Bell at BARC for bioinformatic advice and analysis; N Azubuine and T Nanchung for a constant supply of plates and media.

## Additional information

### Funding

| Funder | Grant reference number | Author |
|---|---|---|
| NIH Office of the Director | DP5 OD017941-01 | David Pincus |
| National Science Foundation | MCB-1350949 | Ahmad S Khalil |
| Alexander and Margaret Stewart Trust | Fellowship | David Pincus |

The funders had no role in study design, data collection and interpretation, or the decision to submit the work for publication.

### Author contributions

XZ, JK, NP, AB, JE, Acquisition of data, Analysis and interpretation of data, Drafting or revising the article; ASK, DP, Conception and design, Acquisition of data, Analysis and interpretation of data, Drafting or revising the article

### Author ORCIDs

David Pincus, http://orcid.org/0000-0002-9651-6858

## Additional files

### Supplementary files

• Supplemental file 1. Yeast strains used in this study.

• Supplemental file 2. Plasmids used in this study.

• Source code 1. This is a .zip file containing 7 .m files to run the heat shock simulations in MATLAB.

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
