## [Decision Letter]

Thank you for submitting your article "Dynamic control of Hsf1 during heat shock by a chaperone switch and phosphorylation" for consideration by *eLife*. Your article has been reviewed by three peer reviewers, including Matthias P Mayer (Reviewer #1) and David S Gross (Reviewer #3), and the evaluation has been overseen by Tony Hunter as the Senior Editor and Reviewing Editor.

The reviewers have discussed the reviews with one another and the Reviewing Editor has drafted this decision to help you prepare a revised submission.

Here, the authors rigorously tested two previously proposed hypotheses for Hsf1-dependent transcriptional regulation of heat shock genes upon temperature upshift in yeast: the chaperone titration model, which assumes that chaperones inhibit Hsf1 under non-stress conditions and are titrated away by stress-denatured proteins, and the Hsf1 phosphorylation model, which argues that Hsf1 is inhibited and activated by differential phosphorylation. They also re-assessed the somewhat enigmatic role of Hsf1 phosphorylation in its transcriptional activation. Through the use of mass spectrometry to identity a large number of heat shock-induced Hsf1 phosphorylation sites, the generation of phosphosite mutants, yeast genetic analysis, mathematical modeling, and a synthetic biology approach, the authors provide convincing evidence that Hsf1 is repressed by the Hsp70 chaperone in the non-stressed state and that titration of Hsp70 by unfolded proteins dominates the initial phase of the heat shock response. Phosphorylation of Hsf1 occurs only at later stages of the response and is responsible for sustained induction of target genes by recruitment of the mediator complex to the heat shock promoters, rather than Hsp70 dissociation.

Essential revisions:

1) There are a number of issues with the modeling of the heat shock response that need addressing:

A) The differential equations mentioned in the Materials and methods seems to contain some mistakes:

First equation: shouldn't it be "- *k*_3_[*HSP][UP*] " instead of "+ *k*_3_[*HSP][UP*]", as one would expect the concentration of free Hsp to decrease as it associates with unfolded protein?

B) Third equation: a "- *k*_6_[*UP*] " parameter could be added to the equation, as the unfolded proteins could also be degraded independently of Hsp70 or in other ways like aggregation be taken out of the equation.

C) How are the initial conditions used for the mathematical model established? Although the model nicely recapitulates the authors' observations, the basic settings of the parameters are far from known values. For example, the number of Hsf1 molecules per yeast cells was determined to be 49 (Chong et al. 2015) to 361 (Kulak et al. 2014), and the number of Ssa1 + Ssa2 molecules together was 19,306 (Chong et al. 2015) to 435,927 (Kulak et al. 2014), which gives a ratio of Hsp70 to Hsf1 of 394 to 1207. In their model the authors use a value of 10. Also, does 10.51 times more unfolded proteins than Hsp70 make sense? Since Hsp70 is considered to be 1% of cellular proteins, such a ratio would mean that 10% of cellular proteins misfold at a temperature when yeast is still able to grow.

D) *k*_1_ and *k*_3_ are bimolecular rate constants and should have the unit min^-1^ M^-1^. The actually observed association rate depends on the concentration of the components, in this case Hsp70, which changes after heat shock.

E) It would also be interesting to look at recovery from heat shock when the unfolded proteins are cleared out and the system returns to its baseline. Does the authors' model also recapitulate the shut-off phase of the heat shock response?

2) The authors present intriguing genetic evidence indicating that Hsp40 enhances the ability of Hsp70 to suppress Hsf1 function (Figure 3). This raises the question of whether Hsp40 enhances Hsp70 binding to Hsf1 in vivo (Figure 1). The authors should compare the co-IP heat shock time course in WT and *ydj1Δ* mutant cells?

3) The prior literature on the interaction of Hsf1 with Hsp70 needs to be discussed in more depth (see below for examples).

A) To strengthen their argument, Guisbert et al. (2013) should be cited, where the Morimoto lab shows through a genome wide screen that knockdown of hsp70 not hsp90 yields the largest activation of heat shock responders, hsp70 and a small hsp.

B) It was shown that deletion of hsc82, the yeast Hsp90, and cpr7, an Hsp90 cochaperone, leads to the induction of the heat shock response (Duina et al. 1998 JBC). The authors should discuss this discrepancy with their inability to find a role for Hsp90.

C) The Shi et al. (1998) paper from the Morimoto group addressing the regulatory mechanism of human HSF1 and deriving similar conclusions to the authors with respect to the role of Hsp70, is relevant, and should be discussed by the authors in the context of their findings.

D) It is becoming more apparent in recent years that Hsf1 has pleotropic effects in cell growth, proliferation, protein translation and stress protection which appear independent of chaperone activation (e.g. Baird et al., Science 2014; Mendillo et al., Cell 2013; Santagata et al., Science 2013). Because Hsf1 has a role in so many cellular processes, some discussion of the possibility that the highly mutagenized Hsf1 protein may not totally reflect wild type Hsf1 function with respect to chaperone activation but still may retain some basal function required for cell viability is needed.

In this regard, although not essential, further experimental evidence that the highly mutated non-phosphorylatable Hsf1 is not acting as a neomorph would strengthen the paper, as would some experiments with another non-amyloidogenic unfolded protein.

---

## [Author Response]

*[…] Essential revisions:*

*1) There are a number of issues with the modeling of the heat shock response that need addressing:*

*A) The differential equations mentioned in the Materials and methods seems to contain some mistakes:*

*First equation: shouldn't it be "- k_3_[HSP][UP] " instead of "+ k_3_[HSP][UP]", as one would expect the concentration of free Hsp to decrease as it associates with unfolded protein?*

We have corrected this typo in the Materials and methods.

*B) Third equation: a "- k_6_[UP] " parameter could be added to the equation, as the unfolded proteins could also be degraded independently of Hsp70 or in other ways like aggregation be taken out of the equation.*

We agree that there are likely to be Hsp70-independent mechanisms of protein degradation and that we could include such a term in the model (just as there are many other chaperones that participate in protein folding and are part of the Hsf1 target regulon). However, since we don’t know the rate at which this occurs relative to Hsp70-dependent degradation, we chose to keep the model narrowly focused on the roles of Hsp70 and omit this term. In future iterations of the model where the aim is to more realistically capture the full scope of events and activities that occur during heat shock, we will factor in other degradation pathways.

*C) How are the initial conditions used for the mathematical model established? Although the model nicely recapitulates the authors' observations, the basic settings of the parameters are far from known values. For example, the number of Hsf1 molecules per yeast cells was determined to be 49 (Chong et al. 2015) to 361 (Kulak et al. 2014), and the number of Ssa1 + Ssa2 molecules together was 19,306 (Chong et al. 2015) to 435,927 (Kulak et al. 2014), which gives a ratio of Hsp70 to Hsf1 of 394 to 1207. In their model the authors use a value of 10. Also, does 10.51 times more unfolded proteins than Hsp70 make sense? Since Hsp70 is considered to be 1% of cellular proteins, such a ratio would mean that 10% of cellular proteins misfold at a temperature when yeast is still able to grow.*

We have explored more thoroughly how the values chosen for the initial conditions affect the model. In particular, we increased the ratio of Hsp70:Hsf1 to a more realistic 500:1and found that we can still get the model to recapitulate the data. We replaced all model plots in the paper with results from simulations that include this constraint. As for the level of unfolded proteins, the 10.51 value is the initial bolus of unfolded proteins that appear immediately upon temperature upshift: this value drops back to zero once homeostasis is restored, consistent with a resumption of cell growth.

*D) k_1_ and k_3_ are bimolecular rate constants and should have the unit min^-1^ M^-1^. The actually observed association rate depends on the concentration of the components, in this case Hsp70, which changes after heat shock.*

We have corrected the units of these terms to reflect that they are bimolecular rate constants. However, we used arbitrary concentration units in the model (rather than molarity), so the units are min^-1^ a.u.^-1^.

*E) It would also be interesting to look at recovery from heat shock when the unfolded proteins are cleared out and the system returns to its baseline. Does the authors' model also recapitulate the shut-off phase of the heat shock response?*

This is a very interesting idea. However, all of the experiments we performed were step inputs where we increased the temperature and maintained it at the elevated level for the duration of the experiment; we did not perform any experiments where we heat shocked the cells and then returned them to the basal temperature to monitor deactivation. Thus we have nothing for the model to recapitulate. We will incorporate this into future work.

*2) The authors present intriguing genetic evidence indicating that Hsp40 enhances the ability of Hsp70 to suppress Hsf1 function (Figure 3). This raises the question of whether Hsp40 enhances Hsp70 binding to Hsf1* in vivo *(Figure 1). The authors should compare the co-IP heat shock time course in WT and ydj1Δ mutant cells?*

We knocked out *YDJ1* in the background of our Hsf1-3xFLAG-V5 strain and performed the co-IP heat shock time course. We also monitored Hsf1 transcriptional activity in the *ydj1∆* strain under basal and heat shock conditions using the HSE-YFP reporter. These data are included in Figure 3—figure supplement 2 and described in the subsection “Hsp70 and Hsp40 suppress Hsf1 overexpression”. We observed that Hsf1 was still able to bind to Hsp70 in the absence of Ydj1 under basal conditions and transiently dissociate during heat shock. In fact, there appears to be more Hsp70 relative to Hsf1 in the *ydj1∆* strain compared to wild type. Also, the degree of Hsp70 dissociation was reduced and the duration of the dissociation was shorter. So at a first pass, it appears that loss of this Hsp40 *increases* the association between Hsf1 and Hsp70 rather than decreasing the association as would be expected if it enhanced binding. However, the problem with interpreting this experiment is revealed by the HSE-YFP reporter assay: the *ydj1∆* cells have ~3-fold higher HSE-YFP levels than wild type cells under basal conditions. Since all four *SSA* genes are Hsf1 targets, it is likely that there is more total Hsp70 in the cell under basal conditions in the *ydj1∆* strain. Thus, without a conditional depletion of Ydj1, the altered initial conditions in the *ydj1∆* strain preclude a direct comparison of the role of Hsp40 in the binding of Hsp70 to Hsf1. We look forward to exploring the roles of the Hsp70 co-chaperones in regulating Hsf1 in depth in future work.

3) The prior literature on the interaction of Hsf1 with Hsp70 needs to be discussed in more depth (see below for examples).

*A) To strengthen their argument, Guisbert et al. (2013) should be cited, where the Morimoto lab shows through a genome wide screen that knockdown of hsp70 not hsp90 yields the largest activation of heat shock responders, hsp70 and a small hsp.*

We have referenced and included discussion of this paper (subsection “Hsp70 and Hsp40 suppress Hsf1 overexpression”).

*B) It was shown that deletion of hsc82, the yeast Hsp90, and cpr7, an Hsp90 cochaperone, leads to the induction of the heat shock response (Duina et al. 1998 JBC). The authors should discuss this discrepancy with their inability to find a role for Hsp90.*

We have included discussion of this discrepancy (Discussion, third paragraph). We can rationalize this genetic result by supposing that loss of Hsp90 function impairs general proteostasis, leading to the accumulation of unfolded proteins that titrate Hsp70 away from Hsf1. We would also invoke this reasoning to explain why Hsp90 inhibitors activate Hsf1.

*C) The Shi et al. (1998) paper from the Morimoto group addressing the regulatory mechanism of human HSF1 and deriving similar conclusions to the authors with respect to the role of Hsp70, is relevant, and should be discussed by the authors in the context of their findings.*

We have now included a discussion of this work with respect to our results in addition to the reference we had already included (subsection “Hsp70 and Hsp40 suppress Hsf1 overexpression” and Discussion, second paragraph).

*D) It is becoming more apparent in recent years that Hsf1 has pleotropic effects in cell growth, proliferation, protein translation and stress protection which appear independent of chaperone activation (e.g. Baird et al., Science 2014; Mendillo et al., Cell 2013; Santagata et al., Science 2013). Because Hsf1 has a role in so many cellular processes, some discussion of the possibility that the highly mutagenized Hsf1 protein may not totally reflect wild type Hsf1 function with respect to chaperone activation but still may retain some basal function required for cell viability is needed.*

In another study from the Pincus and Denic labs (Solís et al. 2016), we have shown that the essential function of yeast Hsf1 is in fact to drive basal expression of chaperones. Thus, the highly mutagenized Hsf1 mutants we describe here, which support growth when expressed as the only copy of Hsf1 in the cell, are performing their essential basal function of driving chaperone gene expression (as can be seen in the RNA seq data in Figure 4).

*In this regard, although not essential, further experimental evidence that the highly mutated non-phosphorylatable Hsf1 is not acting as a neomorph would strengthen the paper, as would some experiments with another non-amyloidogenic unfolded protein.*

To strengthen the argument that Hsf1^∆po4^ is not a neomorph, we monitored its subcellular localization and performed ChIP-seq of wild type Hsf1 and Hsf1^∆po4^. As can be seen in Figure 4—figure supplement 4D, the localization patterns are the same. And importantly, as shown in Figure 4—figure supplement 4E, the genome-wide DNA binding profiles of wild type Hsf1 and Hsf1^∆po4^ are superimposable, indicating that the mutations we introduced in the DNA binding domain do not alter DNA binding site preference. Moreover, the genome-wide mRNA levels in Hsf1^∆po4^ cells matches the wild type to a striking degree (R^2^=0.98, Figure 4). Understanding how these mutants function in response to other challenges to proteostasis, such as over-expression of an unfolded protein and treatment with amino acid analogs such as canavanine or AZC – as well as other stresses that Hsf1 responds to such as glucose depletion and oxidative stress – will be an important focus of future studies.